# Hsf1 and the molecular chaperone Hsp90 support a 'rewiring stress response' leading to an adaptive cell size increase in chronic stress

Samarpan Maiti[1], Kaushik Bhattacharya[1], Diana Wider[1], Dina Hany[1,2], Olesya Panasenko[3], Lilia Bernasconi[1], Nicolas Hulo[4], Didier Picard[1]*

[1]Département de Biologie Moléculaire et Cellulaire, Université de Genève, Genève, Switzerland; [2]On leave from: Department of Pharmacology and Therapeutics, Faculty of Pharmacy, Pharos University in Alexandria, Alexandria, Egypt; [3]BioCode: RNA to Proteins Core Facility, Département de Microbiologie et Médecine Moléculaire, Faculté de Médecine, Université de Genève, Genève, Switzerland; [4]Institute of Genetics and Genomics of Geneva, Université de Genève, Genève, Switzerland

**\*For correspondence:**
didier.picard@unige.ch

**Competing interest:** The authors declare that no competing interests exist.

**Abstract** Cells are exposed to a wide variety of internal and external stresses. Although many studies have focused on cellular responses to acute and severe stresses, little is known about how cellular systems adapt to sublethal chronic stresses. Using mammalian cells in culture, we discovered that they adapt to chronic mild stresses of up to two weeks, notably proteotoxic stresses such as heat, by increasing their size and translation, thereby scaling the amount of total protein. These adaptations render them more resilient to persistent and subsequent stresses. We demonstrate that Hsf1, well known for its role in acute stress responses, is required for the cell size increase, and that the molecular chaperone Hsp90 is essential for coupling the cell size increase to augmented translation. We term this translational reprogramming the 'rewiring stress response', and propose that this protective process of chronic stress adaptation contributes to the increase in size as cells get older, and that its failure promotes aging.

## eLife assessment

This **important** study describes the coordinated regulation of cellular size and protein translation in response to chronic stress as an adaptive mechanism, termed the 'rewiring stress response' regulated by the heat shock response. The evidence supporting this conclusion is **solid**, utilizing diverse methods to monitor and manipulate cell size and evaluate stress resistance. The study could be strengthened by the inclusion of more experiments focused on defining the mechanistic basis of this coordination and broadening the scope of the specific role of the 'rewiring stress response' across different chronic cellular stresses. This work will be of broad interest to researchers interested in diverse fields including cellular proteostasis, stress-responsive signaling, and aging and senescence.

## Introduction

Stress has shaped the interactions between organisms and their environment since the origin of the first cell (*Kültz, 2020b*). Throughout their lifetime, cells and organisms are exposed to different kinds of environmental and cell-intrinsic stresses such as heat, genotoxic agents, oxidative agents, alterations of external pH, hypoxia, nutrient deficiency, osmotic changes, protein misfolding and aggregation,

and various pathological conditions (*Kristensen et al., 2020*). While intense acute stresses can cause lethal damage, in a physiological environment, cells are mostly exposed to mild and often chronic stresses, so that cells have a chance to adapt (*Bijlsma and Loeschcke, 2005*; *Kültz, 2020a*). For example, body or tissue temperature increases of only a few degrees constitute a mild heat stress at the cellular level (*Mainster et al., 1970*; *Evans et al., 2015*; *van Norren and Vos, 2016*; *Baker et al., 2020*; *Rzechorzek et al., 2022*). Similarly, the ecosystem may contain several pollutants that cause a continuous mild oxidative stress (*Taborsky et al., 2021*). Cells and organisms cannot easily escape such stresses and must therefore adapt.

Because cells and organisms need to live with such lifelong stresses, evolution has endowed them with highly conserved stress response pathways (*Liu et al., 1997*). Heat-shock factor 1 (Hsf1), a transcription factor, is the master regulator of cellular stress management (*Li et al., 2017*). Hsf1 governs a protective transcriptional program known as the heat-shock response (HSR), which involves the selective transcription of stress response proteins known as heat-shock proteins (Hsps), most of which are molecular chaperones (*Lindquist and Craig, 1988*; *Richter et al., 2010*; *Vihervaara and Sistonen, 2014*). Molecular chaperones collectively promote the initial folding of newly made proteins, the refolding of unfolded or misfolded proteins, the dissociation of protein aggregates, and the degradation of terminally misfolded or aggregated, and potentially toxic, proteins (*Hartl et al., 2011*). Overall, this process of maintaining protein homeostasis is known as proteostasis. During acute stress of different kinds, apart from the Hsf1-mediated transcriptional response, eukaryotic cells elicit a specific adaptive stress response, known as the integrated stress response (ISR). The ISR reprograms translation to avoid overloading the proteostasis system (*Persson et al., 2020*). In the ISR, global cap-dependent protein translation is reduced, while the translation of some proteins, needed to deal with the challenge, is specifically enhanced.

Hsps are not only required for cells to respond to stress. Even without stress, cells need a basal level of Hsps for folding nascent polypeptide chains, refolding partially denatured or misfolded proteins, or for their degradation (*Balchin et al., 2016*). Hsps are categorized into different families based on their molecular weights. Among those, Hsp90 is essential for the viability and growth of eukaryotic cells and organisms, and it is a major hub of cellular proteostasis through a large number of client proteins (*Taipale et al., 2010*; *Echeverría et al., 2011*; *Johnson, 2012*; *Fierro-Monti et al., 2013*; *Bhattacharya et al., 2020*; *Bhattacharya and Picard, 2021*). It is one of the most abundant proteins in cells (*Nollen and Morimoto, 2002*; *Mollapour et al., 2010*; *Finka and Goloubinoff, 2013*), and in mammals, a large fraction of it is indeed required during prenatal development, and for tissues and cells (*Bhattacharya et al., 2022*). In mammalian cells, there are two different cytosolic Hsp90 isoforms, which are encoded by two different genes, Hsp90α (encoded by the gene *HSP90AA1*) and Hsp90β (encoded by *HSP90AB1*) (*Sreedhar et al., 2004*). Hsp90α is the more stress-inducible isoform, whereas Hsp90β is more constitutively expressed. There are extensive overlapping functional similarities between the two Hsp90 isoforms, which are 84% identical in humans, but there is also some evidence for isoform-specific functions (*Maiti and Picard, 2022*). However, to what extent mammalian cells require these two isoforms during stress adaptation is not clear.

Cellular stress responses, molecular chaperones, and proteostasis are interconnected with cellular and organismal aging (*Labbadia and Morimoto, 2015*; *Hipp et al., 2019*; *Bhattacharya and Picard, 2021*). Although the process of aging is broadly influenced by genetic, epigenetic, and extrinsic factors, it is increasingly apparent that most of these factors ultimately interface with cellular stress response mechanisms (*Kourtis and Tavernarakis, 2011*; *Vilchez et al., 2014*; *Lu et al., 2020*). One of the hallmarks of aging is cellular senescence (*Childs et al., 2015*; *McHugh and Gil, 2018*; *Calcinotto et al., 2019*), which is characterized by a number of features, including a larger cell size. Earlier studies claimed that, as cells become senescent, they stop dividing. But since cell growth, defined as the addition of cell mass, continues, senescent cells become larger (*Hayflick and Moorhead, 1961*; *Mitsui and Schneider, 1976*; *Adolphe et al., 1983*; *Yang et al., 2011*). That is why cells of older mammalian indiviuals are often two or three times larger than those of younger ones (*Cristofalo and Kritchevsky, 1969*; *Treton and Courtois, 1981*; *Demidenko and Blagosklonny, 2008*; *Mammoto et al., 2019*). Remarkably, recent studies have established that a larger cell size is not a consequence, but rather the cause of senescence. If cells fail to scale the amount of their macromolecules as they become larger, this causes cytoplasmic dilution and induces senescence (*Neurohr et al., 2019*; *Lanz et al., 2022*). The question remains why mammalian cells increase their size to the point of becoming senescent.

There are a number of indications that some cells increase their size in response to external cues or functional needs (*Dhawan et al., 2007*; *Boehlke et al., 2010*; *Hall et al., 2012*; *Samak et al., 2016*). However, it is poorly understood whether the cell size itself is the target of regulation or a byproduct of some other adaptation (*Ginzberg et al., 2015*).

Here, we report that mammalian cells gradually enlarge their size to adapt to chronic mild stress (or 'chronic stress' for short). Whereas the cellular response to acute stress has been extensively characterized (*Richter et al., 2010*; *Somero, 2020*), little is known about how cells adapt to chronic mild stress and to what extent certain Hsps or molecular chaperones are involved. We exposed cells to several chronic stresses to investigate these issues, notably also the role of specific cytosolic Hsp90 isoforms. We discovered that in response to chronic stress cells increase their size in an Hsf1-dependent fashion, and that their adaptation to chronic stress is different from the response to acute stress. Unlike acute stress, which causes a shutdown of global translation to reduce the protein burden, chronic stress induces global translation to increase the amount of total proteins. Hsp90, irrespective of its isoform, supports the increase in translation and, through this adaptation, the cell size increase.

## Results

### A mammalian cell model to study the effects of chronic mild stress

To study how mammalian cells adapt to chronic mild stress, we applied several stressors, such as mild heat shock (HS), hypoxia, the chemical stressor sodium arsenite, the protein misfolding agent L-azetidine-2-carboxylic acid (AZC), and tunicamycin (TM) as stressor of the endoplasmic reticulum (ER), to several cell lines (*Figure 1A*, *Figure 1—figure supplement 1A*). To gage the level of chronic stress, which we would consider mild for a given stressor, we decided that it would be the duration or the intensity, where cell death is ≤10%. To determine the chronic mild HS conditions, we kept all cell lines at different temperatures for seven days. We found the threshold for chronic HS to be 39 °C for HEK293T (HEK) and HCT116 cancer cells, and 40 °C for A549 cancer cells and the normal epithelial cell line RPE1. For hypoxia, we determined that 4 days of hypoxia (1% oxygen) are the appropriate threshold for chronic hypoxic stress. For both sodium arsenite and AZC, 5 µM for 5 days is an appropriate threshold for oxidative and proteotoxic stresses, respectively. For TM, we established the optimal dose at 250 nM for 4–5 days. For all stress treatments, to exclude the additional stress and confounding effects of overcrowding, we optimized the seeding density and the size of the cell culture plate such that cells were never more than 70–75% confluent on the day of the analysis (*Figure 1—figure supplement 1B*) (see Materials and Methods for further details).

### Chronic mild stress causes an increase in cell size

While we were checking cell death by flow cytometry to optimize the threshold for the different chronic stresses, we noticed that in all types of chronic stresses, cells increased their size over time (*Figure 1B*, *Figure 1—figure supplement 1C*), as indicated by the forward scatter (FSC) intensity, which is proportional to the diameter (d) of the cell (*Tzur et al., 2011*). Measuring the cell diameter by microscopy (*Figure 1—figure supplement 1D*) confirmed the size increase suggested by the FSC values after 7 days of mild HS. Even though FSC values cannot give absolute numbers, they allow relative changes to be determined and are a reasonable semi-quantitative proxy for cell diameter. Assuming cells in suspension are a round ball, note that a 10% increase in cell diameter translates to more than a 30% increase in cell volume since volume = $(4/3) \times \pi \times (d/2)^3$. It is well supported that mammalian cells control their size via modulation of the cell cycle (*Ginzberg et al., 2015*; *Miettinen et al., 2017*; *Varsano et al., 2017*; *Cadart et al., 2018*). Specifically, a lengthening of the G1 phase is responsible for cell size increases. Hence, we determined whether this increase in cell size is caused by a cell cycle arrest. After seven days of chronic HS, we checked the cell cycle profiles of HEK and A549 cells (*Figure 1C*). Interestingly, the two cell lines showed different cell cycle patterns in chronic HS. While A549 cells have a slightly increased G1 population, HEK cells maintain a similar cell cycle profile. We repeated the analyses at different time points of chronic HS and found that at the initiation of chronic stress, there is a substantial number of cells in the G1 population for both cell lines (*Figure 1D*). Over time, the cell cycle stabilizes in a cell line-specific fashion, suggesting that cells are able to adapt to chronic HS. Importantly, despite the stabilization of the cell cycle during continued chronic stress, cells maintain an enlarged cell size (*Figure 1E*) with an almost equal proliferation rate

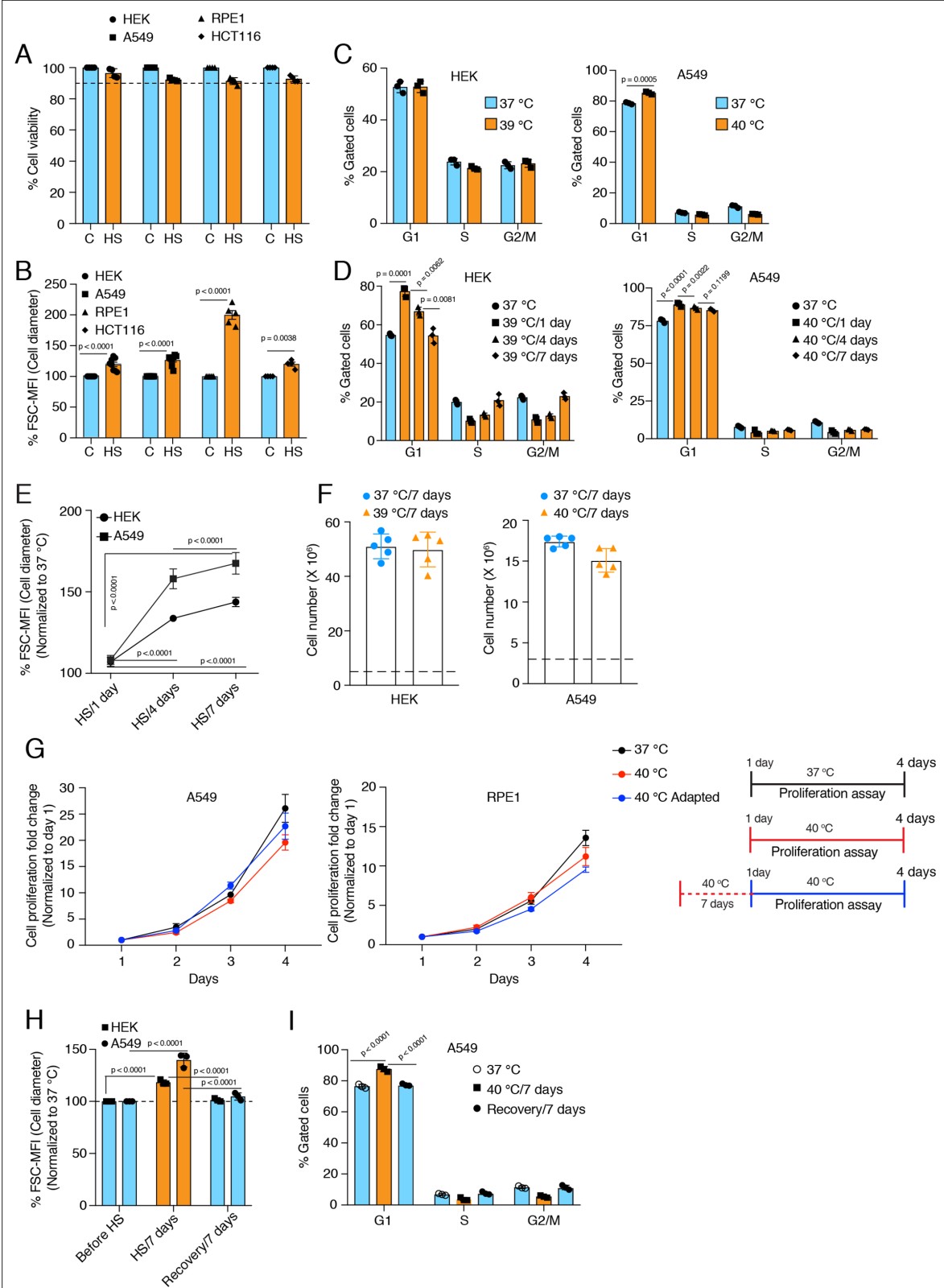

**Figure 1.** Cells increase their size in response to chronic stress. (**A**) Flow cytometric quantification of cell viability under chronic HS for 7 days (HS = 39 °C for HEK and HCT116 cells; HS = 40 °C for A549 and RPE1 cells) (n = 4 biologically independent samples). (**B**) Flow cytometric quantification of cell size after 7 days of chronic HS (biologically independent samples: n = 6 for HEK and A549; n=5 for RPE1; n = 4 for HCT116). (**C and D**) Flow cytometric analysis of cell cycle (n=3 biologically independent samples). (**E**) Flow cytometric quantification of cell size during different time intervals of chronic HS

*Figure 1 continued on next page*

*Figure 1 continued*

(n=4 biologically independent samples). (**F**) Proliferation of HEK and A549 cells at the indicated temperature for the indicated period presented as cell numbers. Cells were seeded at a density of 5x10⁶ and 3x10⁶ per 15 cm plate for HEK and A549 cells, respectively. The numbers of live cells counted after 7 days are plotted (n=5 biologically independent experiments). (**G**) Proliferation of A549 and RPE1 cells measured with a crystal violet assay (n=3 biologically independent experiments). The adapted cells were maintained at 40 °C for 1 week before this experimental start point and continued at 40 °C during the experiment. See scheme of the experiment on the right. Note that the data are normalized to cell numbers on day 1. (**H**) Flow cytometric quantification of cell size in chronic HS and recovery (n=3 biologically independent samples). (**I**) Flow cytometric analysis of cell cycle in chronic HS and post HS recovery (n=3 biologically independent samples). The data are represented as mean values ± SEM for all bar and line graphs. The statistical significance between the groups was analyzed by two-tailed unpaired Student's t-tests.

The online version of this article includes the following source data and figure supplement(s) for figure 1:

**Source data 1.** Values related to all graphs.

**Figure supplement 1.** Cells increase their size in response to different types of chronic stress.

**Figure supplement 1—source data 1.** Values related to all graphs of *Figure 1—figure supplement 1*.

**Figure supplement 2.** Schematic representation of the flow cytometric strategies for cell size and cell cycle analyses.

(*Figure 1F and G*, *Figure 1—figure supplement 1E*) in the beginning of the stress or after 7 days of stress adaptation. Note that the proliferation rates confirm that our experimental conditions avoid overcrowding. To check if this cell size enlargement is reversible, we put the cells back at 37 °C after 7 days of chronic HS. During this recovery period at 37 °C, both HEK and A549 cells returned to their usual size (*Figure 1H*), and A549 cells also reverted back to a normal cell cycle profile (*Figure 1I*, *Figure 1—figure supplement 1F*).

## A minimal level of Hsp90 is required for chronic stress adaptation

A key question is how cells cope with and adapt to these chronic stresses. A complex network of molecular chaperones and their respective co-chaperones acts as a buffer to the myriad of changes during stress (*Bijlsma and Loeschcke, 2005*; *Richter et al., 2010*; *Horwich, 2014*; *Labbadia and Morimoto, 2015*). The cytosolic Hsp90 isoforms are the most abundant molecular chaperones (*Jakob and Buchner, 1994*; *Mayer and Bukau, 1999*; *Young et al., 2001*; *Picard, 2002*). Since Hsp90 had been found to support the size increase of cardiomyocytes following myocardial infarction (*Tamura et al., 2019*), we wondered whether it plays any role in the stress-induced cell size increase, and if so, which one of the two cytosolic Hsp90 isoforms, that is Hsp90α or Hsp90β (*Maiti and Picard, 2022*). To address this, we used our human Hsp90α knockout (KO) and Hsp90β KO HEK and A549 cells (*Figure 2A*; *Bhattacharya et al., 2022*). Note that total Hsp90 levels are correspondingly reduced since there is very little compensation in expression of one isoform when the other one is absent (*Bhattacharya et al., 2022*), and that combined KOs of all cytosolic isoforms (two in humans) are not viable in eukaryotes. Hereafter we will refer to cells lacking one *or* the other isoform as Hsp90α/β KO cells. We observed that the loss of either one of the two isoforms makes them vulnerable to different chronic stresses (*Figure 2B*, *Figure 2—figure supplement 1A*), in contrast to what has recently been reported to happen with Hsp90α/β KO fibrosarcoma cells, which resist as well as wild-type (WT) cells to acute heat and oxidative stress (*Petrenko et al., 2023*). However, even in the absence of one Hsp90 isoform, cells enlarged their size during chronic HS, and hypoxic and oxidative stress conditions (*Figure 2C and D*, *Figure 2—figure supplement 1B and C*). This suggests that the stress-induced cell size increase is not directly associated with a particular cytosolic Hsp90 isoform. So far, we have only considered overall increases of cell size, but it is known that cell size increases are correlated with increases of the size of the nuclei (*Wu et al., 2022* and therein). We therefore measured the size of nuclei and corresponding cell size increase. In this case, we fixed the cells, stained them for filamentous actin and DNA, and measured the areas of the attached cells and their nuclei. We found that both increase proportionately in cells subjected to chronic HS, and as for the total cell size, this is independent of a specific Hsp90 isoform (*Figure 2E*, *Figure 2—figure supplement 1D*).

## The canonical Hsf1 activity regulates the cell size increase in response to chronic stress

Hsf1 is an evolutionarily conserved transcription factor that mediates the cytoprotective HSR throughout the eukaryotic kingdom (*Anckar and Sistonen, 2011*; *Gomez-Pastor et al., 2018*). It

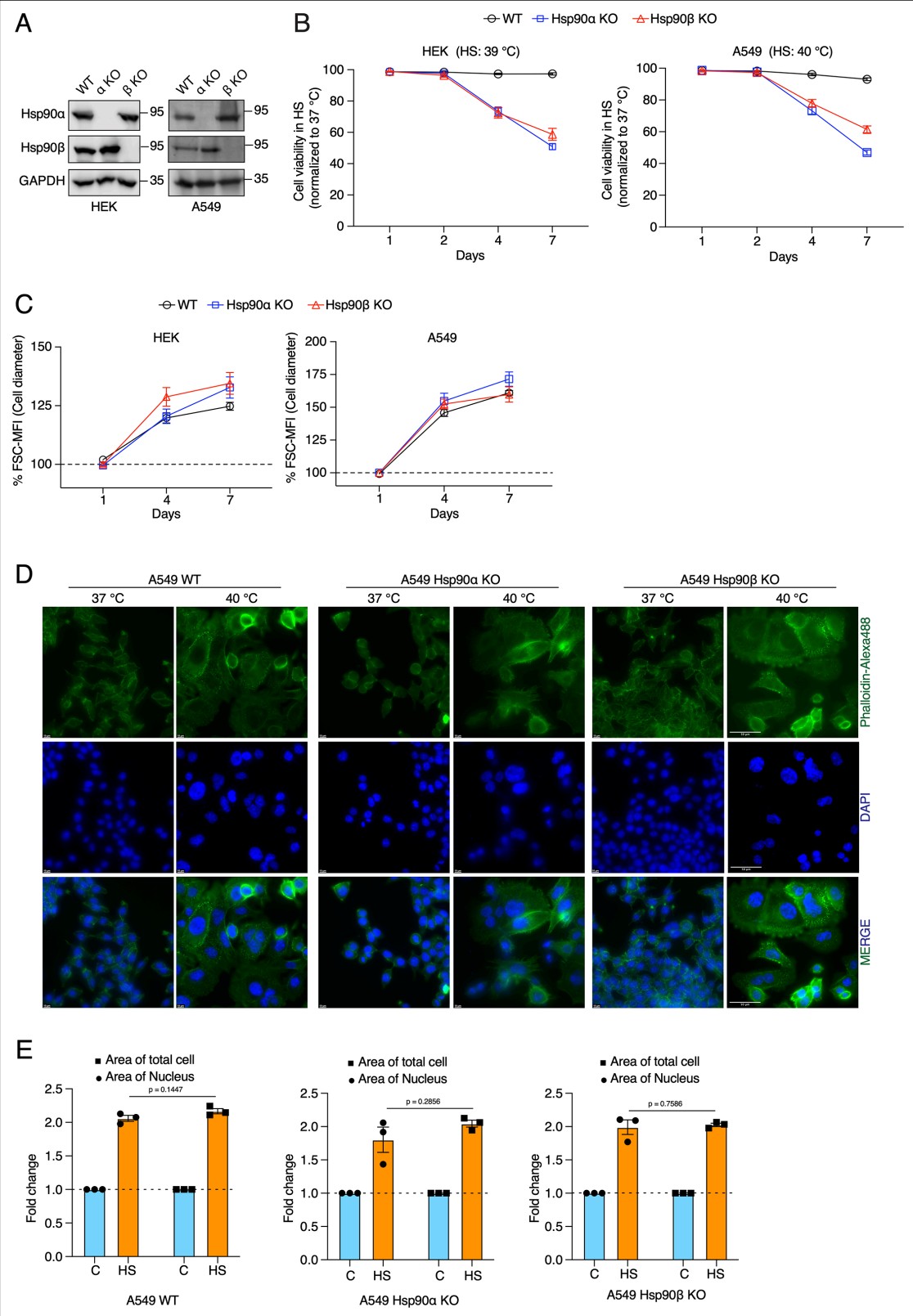

**Figure 2.** Cells increase their overall size and their nuclei, but are unable to adapt to chronic HS in the absence of one of the cytosolic Hsp90 isoforms. (**A**) Immunoblots of Hsp90α and Hsp90β in WT HEK and A549 cells, and their respective Hsp90α/β KO cells. GAPDH serves as the loading control (α KO; Hsp90α KO and β KO; Hsp90β KO) (representative images of n=4 biologically independent experiments). (**B**) Flow cytometric quantification of cell viability of HEK, A549, and their respective Hsp90α/β KO cells in chronic HS at different time points during a period of 7 days (n=5 biologically

*Figure 2 continued on next page*

*Figure 2 continued*

independent samples). Note that the X axis does not have a linear scale and that lines connecting the data points are drawn as a visual aid. (**C**) Flow cytometric quantification of cell size in chronic HS at different time points during a period of 7 days (HS = 39 °C for HEK and 40 °C for A549) (n=4 biologically independent samples) The data are represented as mean values ± SEM for all bar graphs. (**D**) Fluorescence microscopy images of A549 WT and Hsp90α/β KO cells fixed after 4 days of chronic HS. The cytoskeleton is stained with phalloidin-Alexa488 (green), and the nucleus is stained with DAPI (blue). Images were captured with a fluorescence microscope (Zeiss, Germany). The scale bars on the images in the far right column are 50 μM. (**E**) Bar graphs of the area of nuclei and whole cells determined from fluorescent micrographs (representative images are shown in panel D) using ImageJ. We used micrographs from three biologically independent experiments. From each experiment, we measured 30 randomly chosen cells, and their average values were used as one data point. The data for all bar graphs are represented as mean values ± SEM. The statistical significance between the groups was analyzed by two-tailed unpaired Student's t-tests.

The online version of this article includes the following source data and figure supplement(s) for figure 2:

**Source data 1.** Raw and annotated immunoblots.

**Source data 2.** Values related to all graphs.

**Figure supplement 1.** Cells are unable to adapt to chronic stress in the absence of one of the Hsp90 isoforms, but still get larger.

**Figure supplement 1—source data 1.** Values related to all graphs of *Figure 2—figure supplement 1*.

is well established that Hsf1 activity increases in response to acute stresses (*Li et al., 2017*). During stress, mammalian Hsf1 monomers in the cytosol are activated to form trimers, which localize to the nucleus, bind DNA sequences known as heat shock elements (HSE), and trigger the transcription of target genes (*Vihervaara and Sistonen, 2014*). It has been proposed that Hsp90 controls Hsf1 activity by titrating Hsf1 under non-stress conditions (*Zou et al., 1998*; *Leach et al., 2012*; *Lee et al., 2013*; *Hentze et al., 2016*; *Kijima et al., 2018*). Using an Hsf1 reporter plasmid, we found that Hsp90α/β KO cells, and most prominently Hsp90α KO cells, have a higher basal level of Hsf1 activity (*Figure 3A*). In Hsp90α/β KO cells, the higher basal Hsf1 activity could be due to a larger number of Hsp90-free Hsf1 molecules. We did observe that there is more Hsf1 in the nucleus of Hsp90α/β KO HEK cells in non-stressed conditions (*Figure 3B*), and yet Hsf1 becomes even more nuclear in cells of all three genotypes during chronic HS (*Figure 3B*). This also translates to a higher Hsf1 activity, as we could observe with an Hsf1 reporter assay (*Figure 3C*). It is worth pointing out that there are substantial cell-line-specific differences of Hsf1 activity in chronic HS. While A549 cells of all three genotypes show a similarly strong increase, the Hsp90α/β KO HEK cells display significantly stronger stimulation of Hsf1 than WT HEK cells (*Figure 3C*). This suggests a tissue- or cell line-specific role of Hsp90 isoforms or total Hsp90 levels in regulating the Hsf1-mediated heat shock response. To characterize the proteomic changes associated with Hsp90α/β KO and chronic HS, we performed quantitative label-free proteomic analyses of cells maintained in non-stressed conditions, and after one day and four days of HS (*Source data 1*). The proteomic data confirmed the increased basal Hsf1 activity that we had seen in Hsp90α/β KO cells with the Hsf1 reporter assay. Many proteins whose expression was known to be regulated by Hsf1 proved to be upregulated in Hsp90α/β KO cells (*Figure 3—figure supplement 1A* and *Source data 1*). Under chronic HS, cells of all three genotypes were able to increase the expression of several Hsf1 target genes (*Figure 3D*), reminiscent of what we had seen with the Hsf1 reporter assay. Thus, cells are perfectly capable of mounting a HSR in the absence of either one of the two Hsp90 isoforms, suggesting that their increased vulnerability to stress may be due to something else.

To obtain more direct experimental evidence for a role of Hsf1 in the cell size increase induced by chronic HS, we overexpressed WT Hsf1 and a transcriptionally defective Hsf1 mutant (*Kijima et al., 2018*) in HEK, A549, and RPE1 cells. We observed a strong induction of basal Hsf1 activity in cells overexpressing WT Hsf1 but not the mutant form (*Figure 3—figure supplement 1B*), correlating with an increase in cell size under normal non-stress conditions (*Figure 3E*). Taken together, our observations suggest a correlation between the canonical transcriptional Hsf1 activity and cell size. We further strengthened this correlation by determining the effect of inducing Hsf1 activity pharmacologically with capsaicin, which is known to trigger a calcium influx and HSR through the vanilloid receptor TRPV1 (*Hagenacker et al., 2005*; *Bromberg et al., 2013*). We found that A549 cells of all three genotypes display a capsaicin-induced and dose-dependent transcriptional Hsf1 activity (*Figure 3—figure supplement 1C*) and cell size increase (*Figure 3—figure supplement 1D*). Collectively, these results indicate that the transcriptional Hsf1 activity is sufficient to cause a cell size increase. To determine whether Hsf1 is necessary for the cell size increase, we used RNA interference to knock down Hsf1

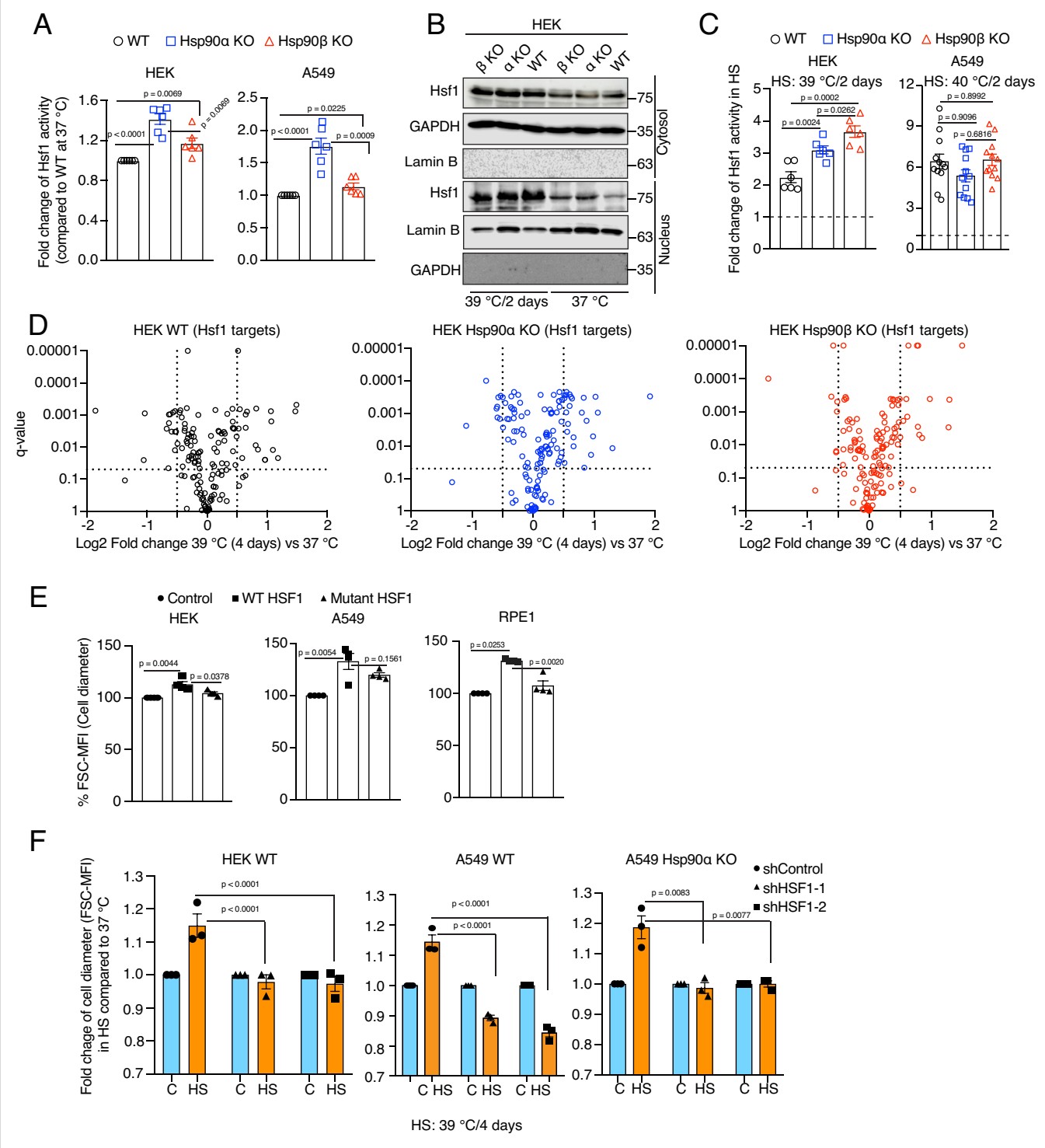

**Figure 3.** Hsf1 regulates cell size in response to stress. (**A**) Fold change of Hsf1 activity of HEK WT, A549 WT, and their respective Hsp90α/β KO cells at 37 °C as measured by luciferase reporter assay (n=3 biologically independent samples and 2 experimental replicates each time). (**B**) Immunoblots of Hsf1 in the cytosolic and nuclear fractions of HEK WT and Hsp90α/β KO cells (α KO, Hsp90αKO; β KO, Hsp90βKO). GAPDH and lamin B1 serve as loading controls (representative blots of n=2 biologically independent experiments). (**C**) Fold change of Hsf1 activity of HEK WT, A549 WT, and their respective Hsp90α/β KO cells in chronic HS as measured by luciferase reporter assay (n=3 biologically independent samples, and 2 experimental replicates each time for HEK; n=3 biologically independent samples, and 4 experimental replicates each time for A549). (**D**) Volcano plots of the normalized fold changes in protein levels of some core Hsf1 target genes (list obtained from https://hsf1base.org/) in chronic HS, determined by quantitative label-free proteomic analysis of Hsp90α/β KO and WT HEK cells. Molecular chaperones, whose expression is regulated by Hsf1, are

*Figure 3 continued on next page*

*Figure 3 continued*

excluded from this dataset. Each genotype was compared with its respective 37 °C control (n=3 biologically independent samples). Log2 fold changes of >0.5 or <–0.5 with q-values (adjusted p-values) of <0.05 were considered significant differences for a particular protein. (**E**) Flow cytometric quantification of cell size of HEK, A549, and RPE1 cells upon overexpression of WT Hsf1 (with plasmid pcDNA-Flag HSF1 wt) or mutant Hsf1 (with plasmid pcDNA-Flag HSF1 C205; retaining only the first 205 amino acids)(**Kijima et al., 2018**), and with plasmid pcDNA3.1(+) as empty vector control. Transfected cells to be measured were identified on the basis of their coexpression of EGFP (n=4 biologically independent experiments). (**F**) Flow cytometric quantification of cell size in chronic HS after knockdown of Hsf1 in HEK WT, A549 WT and Hsp90αKO cells. Here the chronic HS for A549 cells is at 39 °C instead of 40 °C to reduce HS-induced damage in Hsf1 knockdown conditions (n=3 biologically independent samples). For all bar graphs, the statistical significance between the groups was analyzed by two-tailed unpaired Student's t-tests.

The online version of this article includes the following source data and figure supplement(s) for figure 3:

**Source data 1.** Values related to all graphs.

**Source data 2.** Raw and annotated immunoblots.

**Figure supplement 1.** Hsf1 induces cell size in response to stress.

**Figure supplement 1—source data 1.** Values related to all graphs of *Figure 3—figure supplement 1*.

**Figure supplement 1—source data 2.** Raw and annotated immunoblots of *Figure 3—figure supplement 1*.

expression in HEK WT, A549 WT, and Hsp90α KO cells (*Figure 3—figure supplement 1E*). Hsf1 knockdown cells exposed to chronic HS failed to induce Hsf1 activity (*Figure 3—figure supplement 1F*), and could not increase their cell size during adaptation to chronic HS (*Figure 3F*).

So far, we have only shown results obtained with established human cell lines. We also explored the link between Hsp90 and Hsf1 activity in mouse adult fibroblasts (MAFs) established from mice with only a single allele for cytosolic Hsp90 left. These are both homozygous *hsp90α* KO and heterozygous *hsp90β* KO, and will be referred to as 90αKO 90βHET (*Bhattacharya et al., 2022*). These MAFs proved to be substantially bigger than WT MAFs (*Figure 3—figure supplement 1G*), and to display a several-fold higher basal Hsf1 activity (*Figure 3—figure supplement 1H*). It should be emphasized that these MAFs were obtained from adult mice that had escaped the stringent developmental attrition of embryos with this genotype through translational reprogramming of Hsp90β expression (*Bhattacharya et al., 2022*). This may explain why these mutant MAFs, unlike WT MAFs, could not augment their Hsf1 activity nor increase their size any further in chronic HS (*Figure 3—figure supplement 1I and J*). At this point, we conclude that the HSR correlates with cell size during chronic stress, and that both outcomes are mediated by Hsf1.

## Hsp90α/β KO cells overall maintain their chaperome and proteome complexity during chronic stress

Under normal physiological conditions, cells maintain proteostasis through a complex network of molecular chaperones and co-chaperones, a protein collective referred to as the cellular chaperome (*Joshi et al., 2018*; *Yan et al., 2020*). Cellular adaptation to stress does not only require increased Hsf1 activity, but it is also modulated by complex functional relationships between Hsf1 and the cellular chaperome (*Li et al., 2017*). Even though the HSR in chronic stress is not impaired by the loss of one of the Hsp90 isoforms, Hsp90α/β KO cells gradually die off during the chronic HS adaptation period (*Figure 2B*). This raises the question whether the chaperome is compromised in the absence of Hsp90α or β. Using the proteomic data sets mentioned above, we focused on the protein levels of all molecular chaperones and co-chaperones at 37 °C and in chronic HS at different time points. This analysis revealed that Hsp90α/β KO cells could still maintain or increase expression of molecular chaperones and co-chaperones during chronic HS similarly to WT cells (*Figure 4A*, *Figure 4—figure supplement 1A*). Immunoblots confirmed the upregulation of the heat-inducible molecular chaperones Hsp90α (in some instances also Hsp90β), Hsp70, Hsp40, and Hsp27 throughout the chronic stress in cells of all three genotypes (*Figure 4B*, *Figure 4—figure supplement 1B*). This suggests that the chaperome of Hsp90α/β KO cells is not compromised by chronic stress. Similarly, at the whole proteome level, standardized to the same amount of protein, and although there are genotype-specific differences, the proteome remained complex during chronic stress indicating that proteostasis was largely intact (*Figure 4—figure supplement 1C*). Next, we analyzed the Hsp90 interactome. We found that the Hsp90α or Hsp90β KO cells could largely maintain the Hsp90 interactors throughout the chronic stress. These data demonstrate that one Hsp90 isoform is sufficient to support most

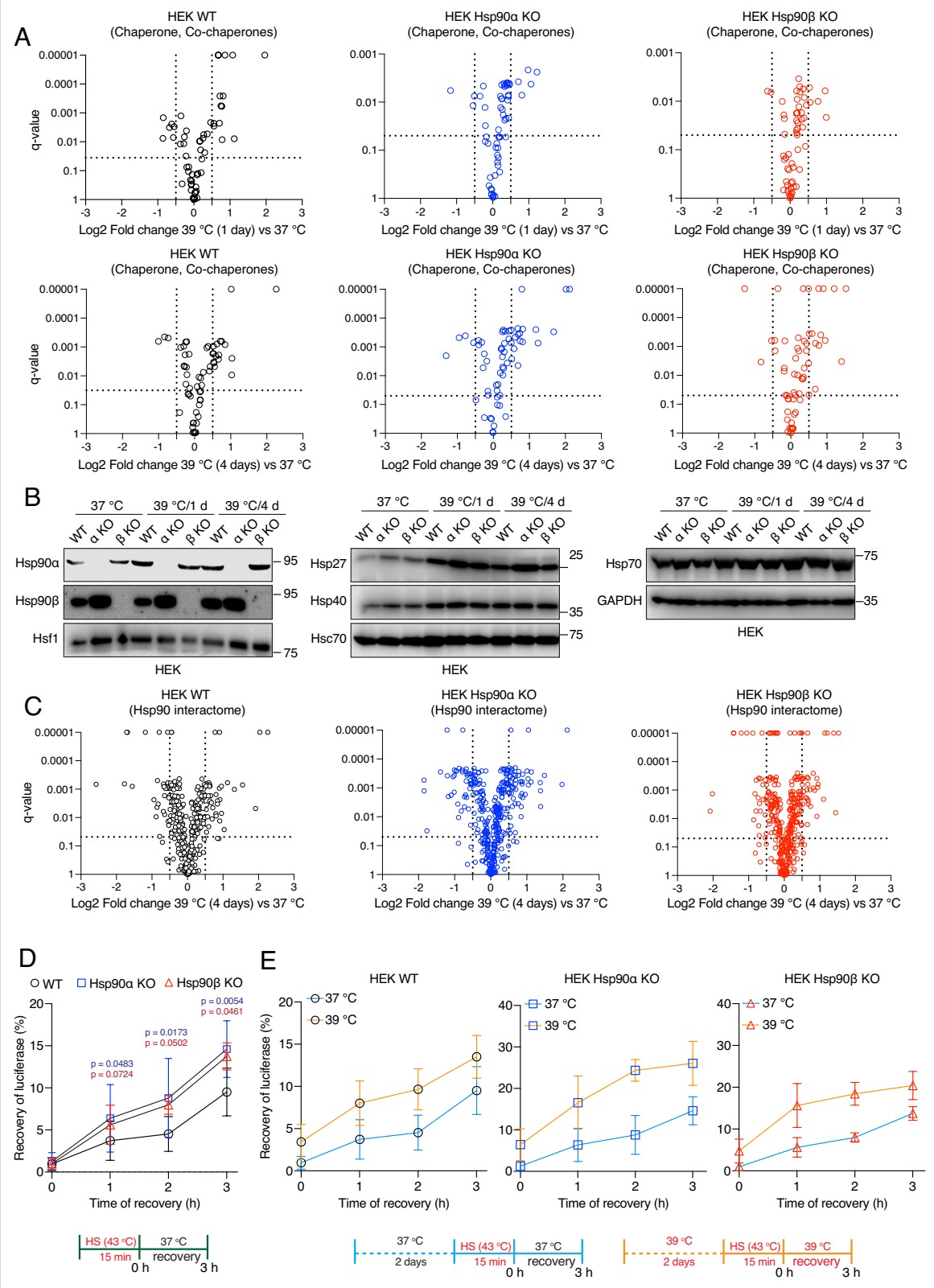

**Figure 4.** Hsp90α/β KO cells maintain chaperones, co-chaperones, and Hsp90 interactors during chronic stress adaptation. (**A**) Volcano plots of the normalized fold changes of molecular chaperones and co-chaperones after 1 and 4 days (first and second rows, respectively) of chronic HS determined by quantitative label-free proteomic analyses of Hsp90α/β KO and WT HEK cells. Each genotype was compared with its respective 37 °C control (n=3 biologically independent samples). Log2 fold changes of >0.5 or <−0.5 with q-values (adjusted p-values) of <0.05 (indicated as stippled lines) were

*Figure 4 continued on next page*

*Figure 4 continued*

considered significant differences for a particular protein. (**B**) Immunoblots of different molecular chaperones in HEK WT and Hsp90α/β KO cells (α KO, Hsp90αKO; β KO, Hsp90βKO). GAPDH serves as the loading control for all three panels (representative of n=2 independent experiments). (**C**) Volcano plots of the normalized fold changes of the Hsp90 interactors (list obtained from https://www.picard.ch/Hsp90Int) after 4 days of chronic HS determined by quantitative label-free proteomic analyses of Hsp90α/β KO and WT HEK cells. Each genotype was compared with its respective 37 °C control (n=3 biologically independent samples). (**D and E**) In vivo refolding of heat-denatured luciferase of control cells (blue line) and cells heat-adapted to 39 °C (orange line). Luciferase activity before the acute HS (at 43 °C) is set to 100% (n=3 biologically independent samples). See scheme of the experiment below. Note the different scales of the Y axes of the bar graphs in panel E. The data are represented as mean values ± SEM for all bar graphs. The statistical significance between the groups was analyzed by two-tailed unpaired Student's t-tests. The p-values for Hsp90α and Hsp90β KO cells are in blue and red, respectively. All p-values are for comparisons to the respective WT.

The online version of this article includes the following source data and figure supplement(s) for figure 4:

**Source data 1.** Raw and annotated immunoblots.

**Source data 2.** Values related to all graphs.

**Figure supplement 1.** Hsp90α/β KO cells maintain molecular chaperones, co-chaperones, and total proteins.

**Figure supplement 1—source data 1.** Raw and annotated immunoblots of *Figure 4—figure supplement 1*.

**Figure supplement 2.** Hsp90α/β KO cells maintain Hsp90 interactors in chronic stress.

interactors (*Figure 4C*, *Figure 4—figure supplement 2*). We conclude from these proteomic analyses that there are no major genotype-specific global changes in levels of chaperones and Hsp90 interactors of cells exposed to chronic HS, but there are some genotype-specific differences that might be worth investigating further in the future.

To address the protein folding ability of Hsp90α/β KO cells more directly, we performed an in vivo refolding assay with exogenously expressed firefly luciferase subjected to unfolding by a short HS at 43 °C. We found that Hsp90α/β KO cells do even better than WT cells under basal conditions (37 °C; *Figure 4D*). The KO cells also showed an increased luciferase refolding ability when they were cultured for 2 days at 39 °C prior to luciferase unfolding (*Figure 4E*). This higher refolding activity of the Hsp90α/β KO cells at 37 °C might be due to the elevated basal activity of Hsf1 (see above) driving higher expression of some other molecular chaperones compared to WT cells (*Figure 4B*). Similarly, Hsp90α/β KO cells might do better than WT cells when subjected to a mild HS for 2 days because of their ability to further increase the levels of other molecular chaperones, such as Hsp27, Hsp40, and Hsp70, during chronic HS. So far, these observations suggest that a reduced level of Hsp90 is at least in part functionally compensated by other molecular chaperones in these assays.

## A normal level of Hsp90 is required to maintain the cytoplasmic protein density in chronic stress

If Hsp90α/β KO cells do as well or even better than WT cells for almost all parameters investigated so far, what causes them to be more sensitive to stress (*Figure 2B*)? Water accounts for about 70% of the weight of a typical cell, with proteins, nucleic acids, lipids, and polysaccharides contributing most of the remaining mass. The largest contribution to cellular dry mass is typically from proteins, followed by nucleic acids and lipids. It is known that when cells grow larger, they must maintain all macromolecules at proportionate levels (*Lloyd, 2013*; *Kempe et al., 2015*; *Lin and Amir, 2018*; *Berenson et al., 2019*; *Lanz et al., 2022*; *Miettinen et al., 2022*). To achieve this for proteins, cells need to increase the total amount of proteins proportionately to their size increase, a process that can be referred to as scaling. Uncoupling of protein synthesis and cellular volume, for example in excessively large cells, causes a dilution of the cytoplasm, which results in cellular senescence and aging (*Neurohr et al., 2019*).

We tested cytoplasmic density by measuring the mobility of EGFP in vivo using fluorescence recovery after photobleaching (FRAP) experiments (*Figure 5—figure supplement 1A*). We calculated the t-half values and diffusion rates (*Persson et al., 2020*), and found that only WT cells could maintain the same cytoplasmic density under chronic HS as at 37 °C (*Figure 5A*, *Figure 5—figure supplement 1B–D*). Hsp90α/β KO cells of both cell lines showed lower t-half values and higher diffusion coefficients in chronic stress indicating that they have a reduced cytoplasmic density (*Figure 5A*, *Figure 5—figure supplement 1B–D*). This observation suggests that a full complement of cytosolic Hsp90 is necessary to maintain the cytoplasmic density with the increase of cell size in chronic stress.

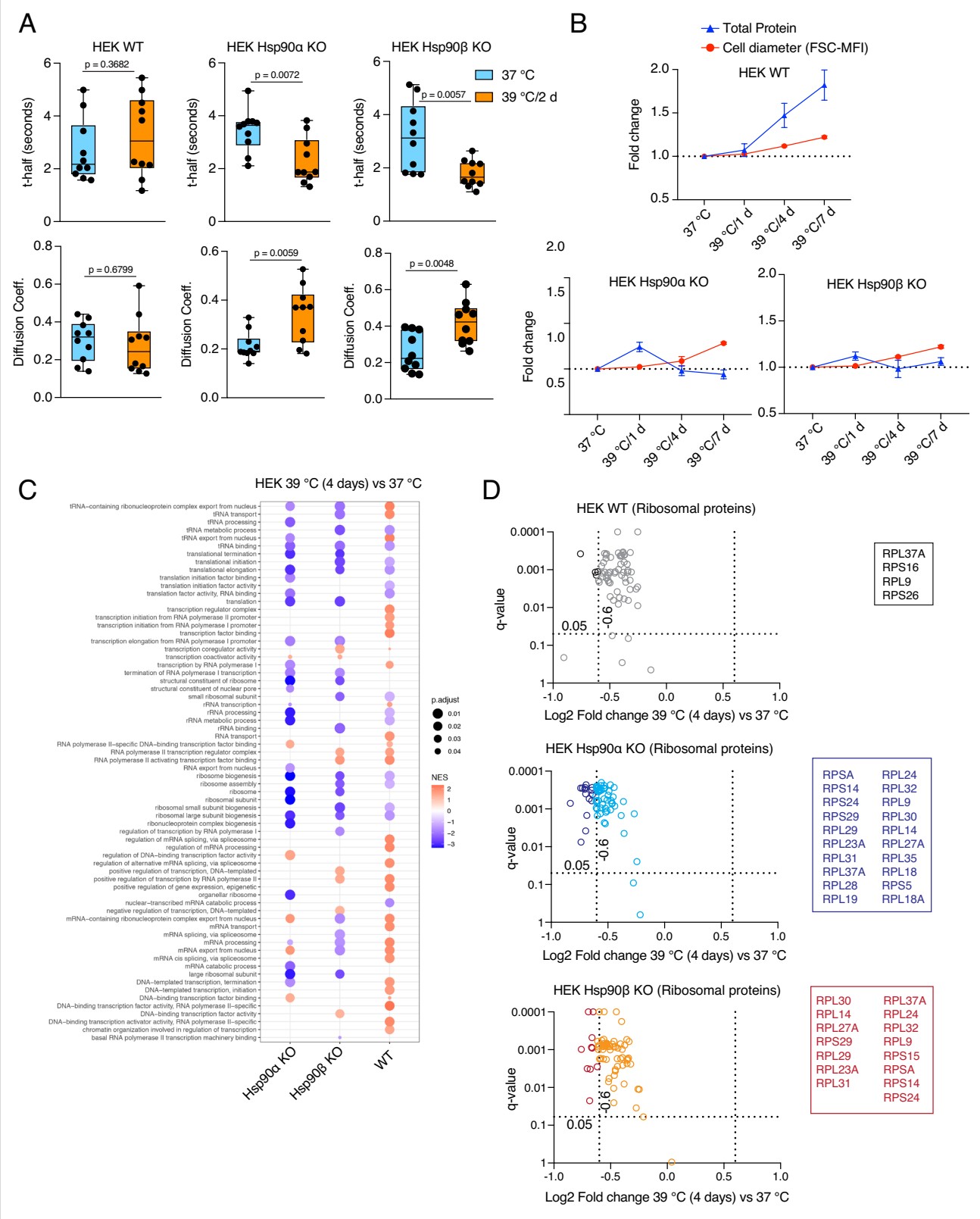

**Figure 5.** Hsp90α/β KO cells suffer from cytoplasmic protein dilution during adaptation to chronic stress. (**A**) FRAP experiments with control and heat-adapted live cells expressing EGFP. The respective box plots show the t-half values of recovery of EGFP fluorescence and the apparent EGFP diffusion coefficients (n=10 cells from two biologically independent experiments). The statistical significance between the groups was analyzed by two-tailed unpaired Student's t-tests. (**B**) Fold change of cell size (represented by the FSC-MFI values) and total proteins (determined as MFI-FL1 values) in chronic

*Figure 5 continued on next page*

*Figure 5 continued*

HS as analyzed by flow cytometry. Cells were fixed, and total proteins were stained using Alexa Fluor 488 NHS ester (n=3 biologically independent experiments). Lines connecting the data points are drawn as a visual aid. (**C**) GSEA plot showing the up- or down-regulation (red and blue, respectively) of pathways associated with cellular transcription and translation after 4 days of chronic HS compared to the respective 37 °C controls; NES, normalized enrichment score. (**D**) Volcano plots of the normalized fold changes of the ribosomal proteins (list obtained from http://ribosome.med.miyazaki-u.ac.jp/) after 4 days of chronic HS determined by quantitative label-free proteomic analysis in Hsp90α/β KO and WT HEK cells. Each genotype was compared with its respective 37 °C control (n=3 biologically independent samples). Log2 fold changes of >0.6 or <–0.6 with q-values (adjusted p-values) of <0.05 (indicated as stippled lines) were considered significant differences for a particular protein. The boxes beside the volcano plots list the corresponding proteins that were significantly downregulated.

The online version of this article includes the following source data and figure supplement(s) for figure 5:

**Source data 1.** Values related to all graphs.

**Figure supplement 1.** Wild-type cells maintain cytoplasmic density and total protein ratio during stress-induced cell size increase.

**Figure supplement 1—source data 1.** Values related to all graphs of *Figure 5—figure supplement 1*.

There is no significant difference between Hsp90α/β KO cells, which indicates that total Hsp90 levels rather than a specific isoform are critical for maintaining the cytoplasmic density.

If the cytosol is more diluted, this should be reflected in the total amount of proteins. Therefore, we collected the same number of non-stressed and stressed enlarged cells after 1 day and 4 days of chronic HS, lysed them in the same amount of lysis buffer, and measured the protein concentrations in the lysates. We found that after 4 days of chronic stress, the total amount of proteins was higher in WT cells, whereas it decreased in Hsp90α/β KO cells (*Figure 5—figure supplement 1E*). Quantitation by flow cytometric analysis of total cellular protein stained with an amine-reactive dye revealed qualitatively similar results. The total amount of proteins per cell increased with cell size in WT but not in Hsp90α/β KO cells (*Figure 5B*, *Figure 5—figure supplement 1E*). These results demonstrate that cells are unable to maintain the ratio of total protein to cell size in chronic stress when Hsp90 levels are reduced.

## Normal levels of Hsp90 are required to scale protein biosynthesis during stress-induced cell size enlargement

One of the key features of the acute stress response of mammalian cells is a global inhibition of translation (*Liu et al., 2013*; *Shalgi et al., 2013*; *Advani and Ivanov, 2019*; *Jobava et al., 2021*). We demonstrated here that WT cells exposed to chronic stress increase the total amount of proteins as they get larger, raising the question of how they do it. Revisiting our proteomic data set, we performed a gene set enrichment analysis (GSEA) and used an output list of enriched gene ontology (GO) terms focused on the pathways related to transcription and translation. Although even WT cells appear to have some deficits associated with the protein translation machinery, Hsp90α/β KO cells are more severely affected (*Figure 5C*). For example, the levels of ribosomal proteins are reduced the least in WT compared to Hsp90α/β KO cells under chronic stress (*Figure 5D*). A reduction of the core machinery of translation may explain why Hsp90α/β KO cells have decreased amounts of total proteins compared to WT in chronic stress. However, it does not explain the observation of the increased amount of total proteins in WT cells under chronic stress. To address this issue, we checked global translation by labelling nascent polypeptide chains with a fluorescent version of puromycin, which allows measurements for individual cells by flow cytometry. We saw a strong increase of labelled nascent polypeptides in WT HEK cells both on day 1 (*Figure 6—figure supplement 1A*) and day 4 (*Figure 6A*) of adaptation to chronic HS. By comparison, Hsp90α/β KO HEK cells failed to maintain the same rate of protein biosynthesis under chronic HS (*Figure 6A*, *Figure 6—figure supplement 1A*). While the stress-adapted bigger cells have a higher level of total translation (*Figure 6A*), it is known that acute HS causes ribosomal pausing of translating mRNAs (*Shalgi et al., 2013*). Polysome profiling of WT cells adapted to stress for four days showed that the polysome profiles of cells grown at 37 °C or in chronic HS are largely similar (*Figure 6B*). Similar peaks of single ribosomal particles (40 S and 60 S), monosomes (80 S), and polysomes (both small and large) suggest that the association of ribosomes and mRNAs in WT cells is not affected during adaptation to chronic stress. This raises the question of whether the ISR is not triggered when the stress is chronic. To address this, we did puromycin labeling at earlier time points of chronic HS. We observed that at the beginning of chronic

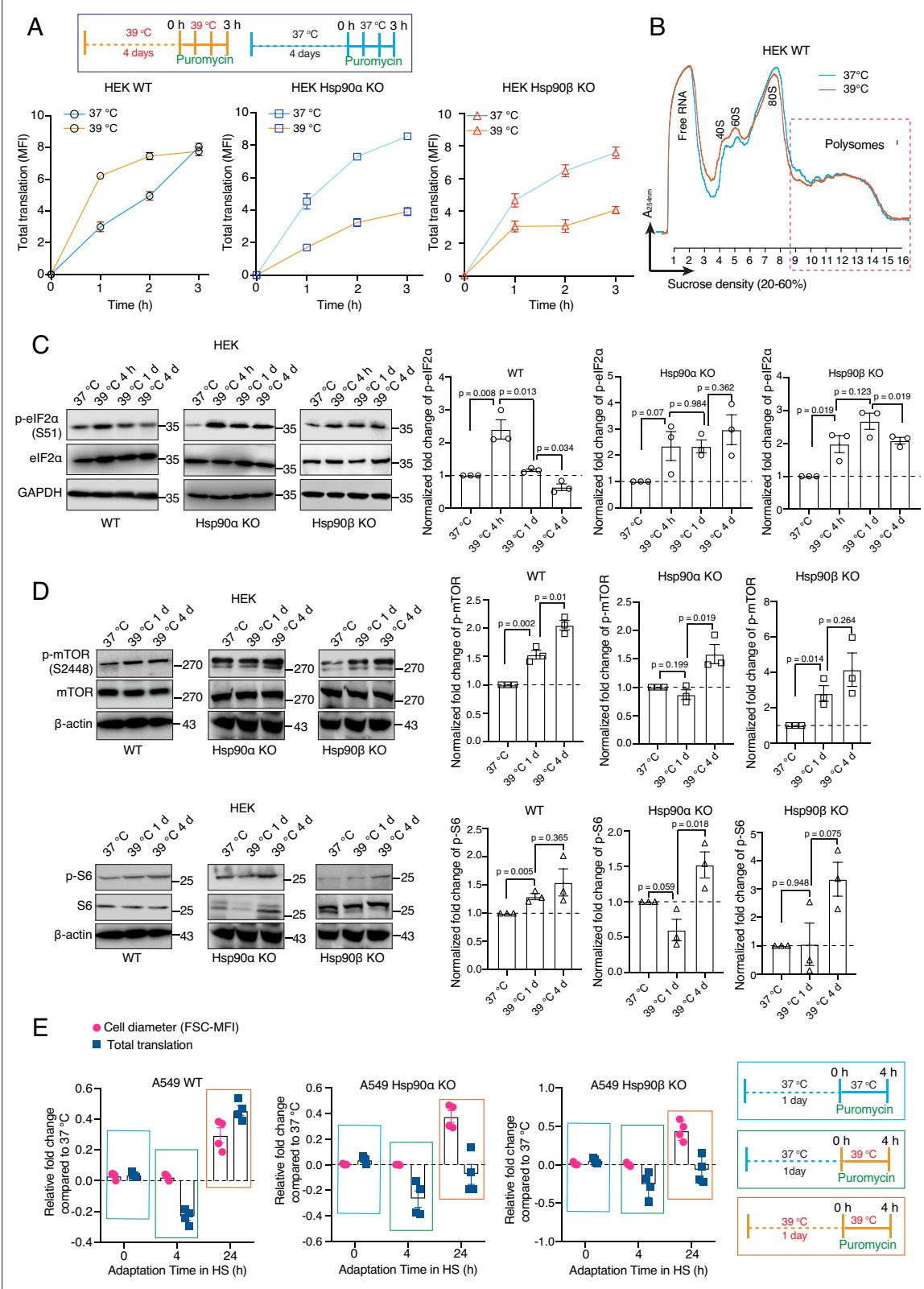

**Figure 6.** Hsp90 is crucial for adapting translation to chronic stress. (**A**) Flow cytometric analysis of total translation of HEK WT and Hsp90α/β KO cells at 37 °C and after 4 days of chronic HS (see scheme of the experiment on the top). Nascent polypeptide chains were labeled with OP-puromycin during cell culture, and the incorporation of puromycin at different time points was analyzed (n=4 experimental samples). (**B**) Representative polysome profiles of HEK WT cells at 37 °C and after 4 days of chronic HS (representative of n=2 biologically independent experiments). (**C and D**) Immunoblots of the

*Figure 6 continued on next page*

*Figure 6 continued*

translation-related proteins eIF2α, mTOR, and S6. GAPDH and β-actin serve as loading controls. Note that the same β-actin immunoblot is used twice as internal loading control since the other immunoblots of panel D are from the same experiment. Representative immunoblots are shown, but the corresponding bar graphs on the right are based on the quantitation of three biological replicates. Values were normalized to the loading controls, and the phosphoproteins were normalized to their respective total protein. (**E**) Relative fold changes of total translation and cell size in the early phase of adaptation to chronic HS (see schemes of experiments on the right) for A549 WT and Hsp90α/β KO cells. The data are represented as mean values ± SEM for all bar graphs. The statistical significance between the groups was analyzed by two-tailed unpaired Student's t-tests.

The online version of this article includes the following source data and figure supplement(s) for figure 6:

**Source data 1.** Values related to all graphs.

**Source data 2.** Raw and annotated immunoblots.

**Figure supplement 1.** Hsp90 requirement for cellular translation during adaptation to chronic stress, and early time points of translational adaptation of wild-type cells.

**Figure supplement 1—source data 1.** Values related to all graphs of *Figure 6—figure supplement 1*.

**Figure supplement 1—source data 2.** Raw and annotated immunoblots of *Figure 6—figure supplement 1*.

**Figure supplement 2.** Differential effects of Hsp90 levels on eIF2α, mTOR, and S6.

**Figure supplement 2—source data 1.** Raw and annotated immunoblots of *Figure 6—figure supplement 2*.

**Figure supplement 2—source data 2.** Values related to all graphs of *Figure 6—figure supplement 2*.

**Figure supplement 3.** Schematic representation of the flow cytometric strategies to measure translation.

stress (4 hr), reminiscent of the ISR (*Persson et al., 2020*), translation is reduced as seen by flow cytometry (*Figure 6—figure supplement 1B*) and puromycin labelling of nascent chains detected by immunoblotting (*Figure 6—figure supplement 1C*). Moreover, both HEK and A549 cells of all three genotypes displayed an increased inhibitory phosphorylation of the eukaryotic translation initiation factor 2 A (eIF2α), a hallmark of the ISR (*Pakos-Zebrucka et al., 2016*; *Wek, 2018*), after 4 hr of HS (*Figure 6C*, *Figure 6—figure supplement 2*). After having been exposed to chronic HS for one day and more, WT cells increase total translation, to an even higher level than the basal translation at 37 °C (*Figure 6—figure supplement 1B*), consistent with the higher translation rate observed by puromycin labeling of nascent chains (*Figure 6A*, *Figure 6—figure supplement 1A*). As cells remained in chronic HS, the phosphorylation of eIF2α only dropped in WT but not Hsp90α/β KO cells (*Figure 6C*, *Figure 6—figure supplement 2*). These findings indicate that at the very beginning of chronic HS, an ISR is induced with the accompanying inhibitory phosphorylation of eIF2α and global reduction of translation. WT cells but not Hsp90α/β KO cells, once adapted to the stress, recover normal or even increased global translation. Translational recovery of WT cells is also reflected in several other translation markers and regulators. After an initial drop in some cases, we could see an increase in the phosphorylation of mTOR and S6 during adaptation to chronic HS (*Figure 6D*, *Figure 6—figure supplement 2*), consistent with earlier reports demonstrating increased phosphorylation mTOR and S6 upon stress (*Kakigi et al., 2011*; *Yoshihara et al., 2013*; *Gao et al., 2015*). In our case, after 4 days of chronic HS, even Hsp90α/β KO cells displayed an increase in mTOR and S6 phosphorylation, suggesting that eIF2α dephosphorylation may constitute a critical (or perhaps *the* critical) and Hsp90-dependent aspect of stress adaptation.

The mTOR complex 1 (mTORC1) is known to regulate cell size by regulating cellular translation (*Fingar et al., 2002*). Hsf1 in a non-transcriptional mode has been linked to regulating organ and cell size by preserving mTORC1 activity (*Su et al., 2016*). And yet, we saw that Hsp90α/β KO cells increase their size in chronic HS despite failing to augment global translation. We therefore determined cell size and translation with A549 cells at different time points following exposure to chronic HS (*Figure 6E*). In contrast to WT cells for which cell size and translation appear to be coupled, Hsp90α/β KO cells increase cell size while translation is still reduced. Thus, cell size and translation must be coupled for adaptation to chronic stress. It remains to be seen how Hsp90α/β KO cells manage to increase their size without a concomitant scaling of translation and to what extent the mTORC1-mediated regulation of translation is involved.

## Hsp90α/β KO cells are efficient at maintaining cellular proteostasis under normal unstressed conditions

In experiments presented above, we showed that the loss of one Hsp90 isoform does not impair the protein refolding ability (*Figure 4D and E*), but reduces translation under chronic stress. Proteolysis is complementary to translation and folding, and crucial to prevent cytotoxicity by eliminating damaged proteins (*Hipp et al., 2019*). Cells promote the degradation of terminally misfolded proteins via the autophagy-lysosomal pathway (ALP) or the ubiquitin-proteasome system (UPS) (*Glickman and Ciechanover, 2002*; *Klaips et al., 2018*). We performed an in vivo activity assay for the UPS, which involved the transient expression and flow cytometric quantitation of a degradation-prone ubiquitin-GFP fusion protein (Ub-R-GFP) and its stable counterpart (Ub-M-GFP) as a control (*Dantuma et al., 2000*). We observed no significant differences in the UPS activity of Hsp90α/β KO cells compared to WT cells at 37 °C (*Figure 7—figure supplement 1A*). We checked the ALP by measuring the autophagic flux using a mCherry-GFP-LC3 reporter (*Leeman et al., 2018*). Here also, we observed that the autophagic flux remains unchanged in Hsp90α/β KO cells (*Figure 7—figure supplement 1B*). These results lead us to conclude that one Hsp90 isoform is sufficient to maintain cellular proteostasis under normal, unstressed conditions.

## A normal level of Hsp90 is required to maintain cellular proteostasis in chronic stress

We recently reported that Hsp90α/β KO HEK cells accumulate insoluble proteins in long-term mild HS (*Bhattacharya et al., 2022*). We therefore checked the above-mentioned cellular proteostasis axes after two days of adaptation to chronic stress. We found that the absence of either one of the two Hsp90 isoforms causes a deficit in UPS activity in vivo (*Figure 7A*) and autophagic flux (*Figure 7B*). Hsp90α/β KO cells subjected to chronic HS also had reduced proteasomal activity when measured in vitro (*Figure 7C*). When we expressed the aggregation-prone model protein EGFP-Q74 (*Narain et al., 1999*) in stress-adapted Hsp90α/β KO cells, more and larger aggregates of EGFP-Q74 were readily detectable (*Figure 7D*, *Figure 7—figure supplement 1C*). This suggests that Hsp90α/β KO cells cannot efficiently maintain proteostasis in chronic stress. Since Hsp90α KO and Hsp90β KO cells are largely affected the same way, we conclude that there is no isoform specificity for maintaining proteostasis during stress, but that total Hsp90 levels above a certain threshold might be the critical parameter.

## Enlarged cells are more resistant to subsequent stress

We hypothesized that the stress-induced cell size enlargement is a protective adaptation. To address this, we induced the enlargement or reduction of cell size with different inhibitors and examined how cell size affects cell survivability (*Figure 8A*). At first, we enlarged the A549 WT and Hsp90α/β KO cells by treating them with a CDK4/6 inhibitor (*Figure 8B*), which is known to promote a G1 arrest-associated increase in cell size (*Neurohr et al., 2019*). In contrast to what we had seen with chronic HS, upon becoming bigger in response to treatment with the CDK4/6 inhibitor, Hsp90α/β KO cells were able to scale the amount of total proteins (*Figure 8—figure supplement 1A and B*). This demonstrates that scaling total protein is possible even in the absence of one Hsp90 isoform and a reduced amount of total Hsp90 when cells are not in chronic HS. We showed above that Hsp90α/β KO cells were dying in chronic HS, possibly because they could not scale their protein with increasing cell size. We therefore wondered whether Hsp90α/β KO cells, enlarged with prior treatment with the CDK4/6 inhibitor, might be more resistant to chronic HS because they would already have scaled their total protein. After treatment with CDK4/6 inhibitors for three days, we washed off the inhibitors and subjected the cells to chronic HS for three days. We observed that cells that were already enlarged due to the CDK4/6 inhibitor treatment did not get even bigger in chronic HS. As predicted, these pre-enlarged Hsp90α/β KO cells were more resistant and less prone to apoptosis and cell death (*Figure 8B*). Thus, when given a chance to get bigger with protein scaling, even cells lacking one Hsp90 isoform are more resistant to chronic stress.

We further explored the connection between increase in cell size and scaled protein, and stress resistance with other stresses and cell lines. We treated the cells pre-enlarged with the CDK4/6 inhibitor with an acutely toxic concentration of sodium arsenite for one day. Again, pre-enlarged A549 cells of all three genotypes and the normal RPE1 cells were mo*re resistant to this subsequent stress*

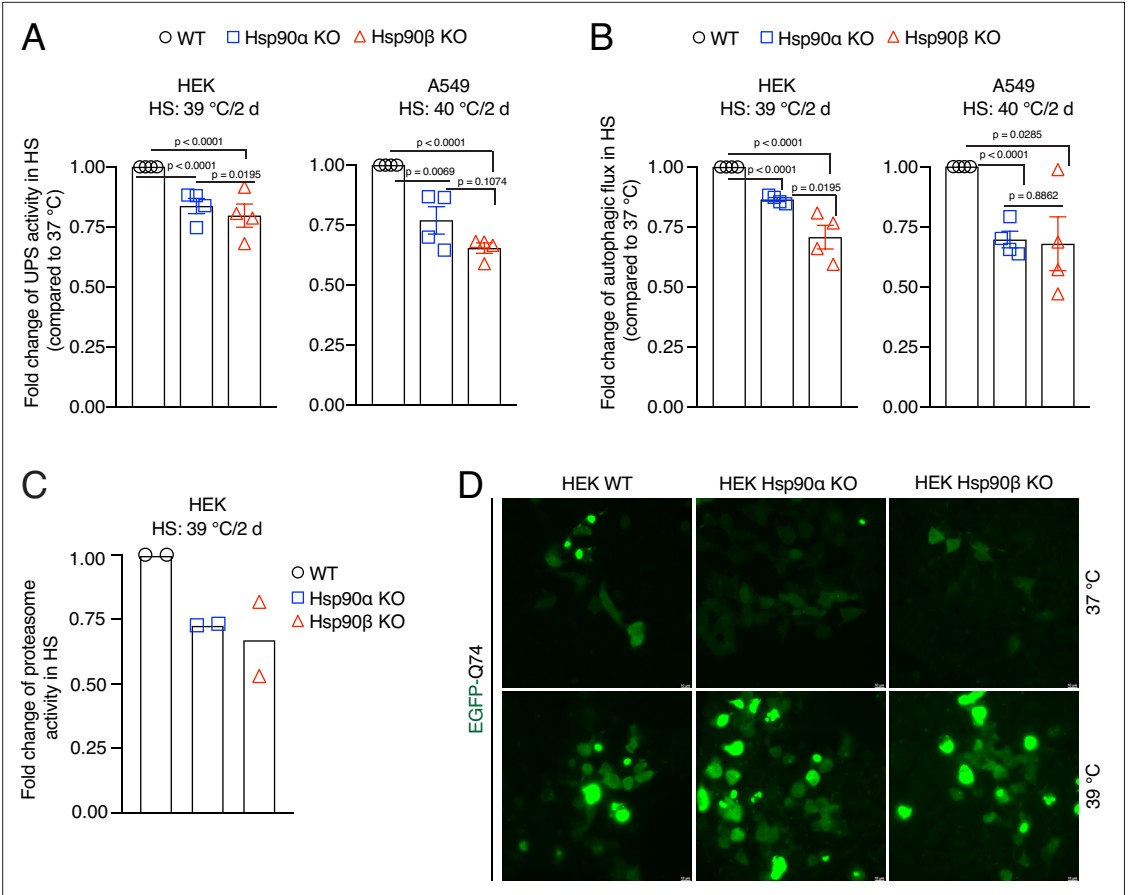

**Figure 7.** Hsp90 is crucial for cellular proteostasis during adaptation to chronic stress. (**A**) Flow cytometric determination of the in vivo UPS activity in chronic HS compared to 37 °C, using the Ub-M-GFP and Ub-R-GFP reporter proteins (n=4 biologically independent samples). (**B**) Flow cytometric measurement of autophagic flux in chronic HS compared to 37 °C, using a mCherry-GFP-LC3 reporter. Flux is calculated as the ratio of the mean fluorescence intensities of mCherry and GFP-positive cells (n=4 biologically independent samples). (**C**) In vitro steady-state proteasomal activity with lysates of HEK WT, and Hsp90α and Hsp90β KO cells determined by measuring fluorescence of the cleaved substrate suc-LLVY-AMC (n=2 biologically independent samples). (**D**) Fluorescence micrographs of cells expressing the fusion protein EGFP-Q74 visible as aggregates with green fluorescence. The scale bars in the zoomable micrographs indicate 10 μm (images are representative of n=2 independent biological samples). The data are represented as mean values ± SEM for all bar graphs. The statistical significance between the groups was analyzed by two-tailed unpaired Student's t-tests.

The online version of this article includes the following source data and figure supplement(s) for figure 7:

**Source data 1.** Values related to all graphs.

**Figure supplement 1.** Hsp90α/β KO cells maintain WT levels of protein degradation activities in unstressed conditions, but have more protein aggregates.

**Figure supplement 1—source data 1.** Values related to all graphs of *Figure 7—figure supplement 1*.

**Figure supplement 2.** Schematic representation of the flow cytometric strategies for measuring autophagic flux and in vivo UPS activities.

---

(*Figure 8—figure supplement 1C*). We then wondered whether reducing cell size would have the opposite effect. We treated HEK and A549 WT, and Hsp90α/β KO cells with rapamycin to inhibit mTOR-induced cell growth (*Fingar et al., 2002*; *Figure 8C*, *Figure 8—figure supplement 1D*). Cell size was indeed reduced after three days of rapamycin treatment (*Figure 8C*, *Figure 8—figure supplement 1D*). Surprisingly, we found that even in the presence of rapamycin cells enlarged their size under chronic HS, albeit not up to the level of cells without rapamycin pretreatment. More of these comparatively smaller rapamycin-treated WT cells were apoptotic than of the bigger cells not treated with rapamycin (*Figure 8C*, *Figure 8—figure supplement 1D*). The increased cell death could also be due to the translational inhibition, in light of the fact that WT cells display increased total translation and mTOR activity to adapt to chronic stress (*Figure 6D*, *Figure 6—figure supplement 2*).

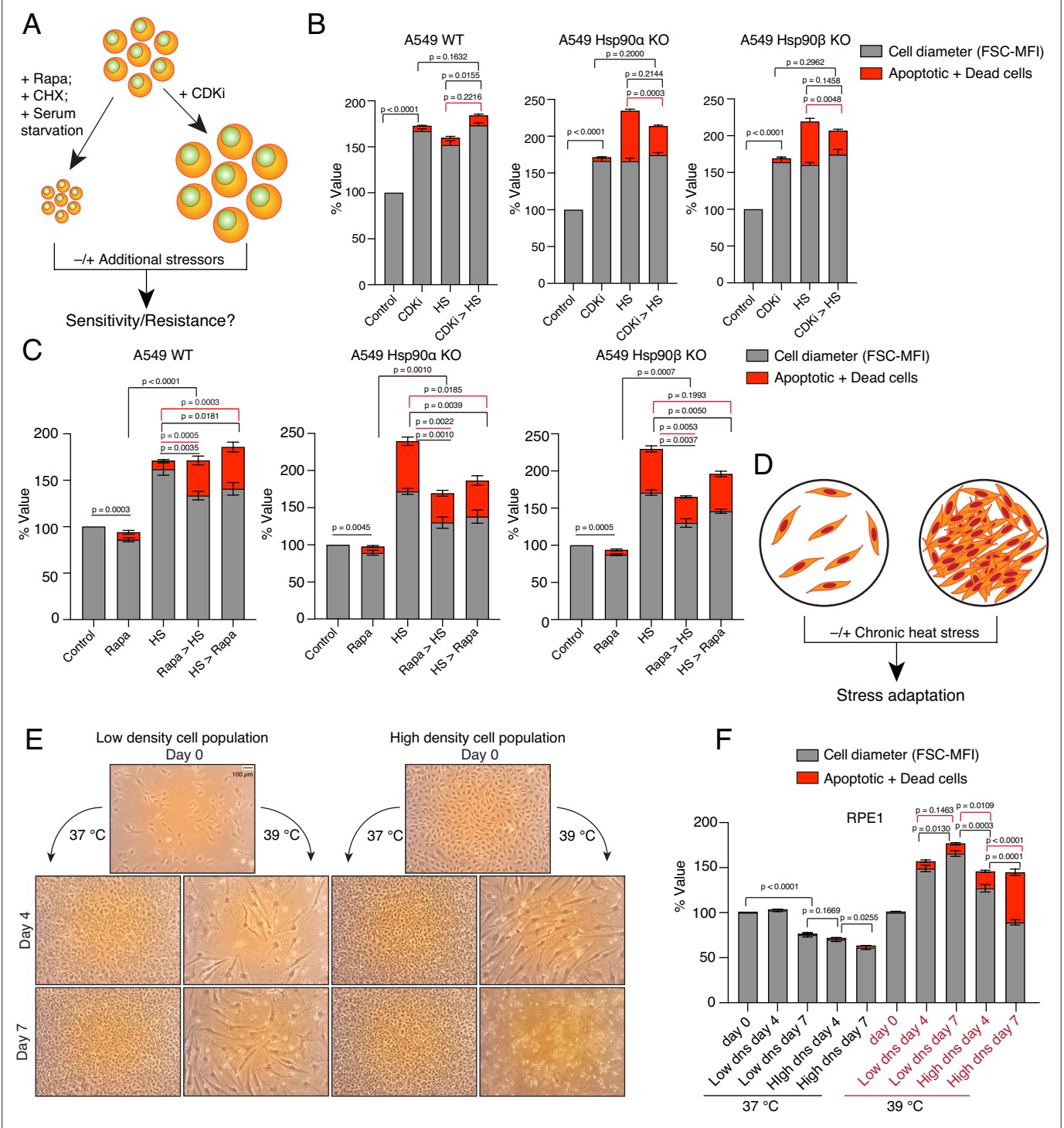

**Figure 8.** Enlarged cells are more resistant to additional stress. (**A**) Scheme of cell size enlargement or reduction experiments. CHX, cycloheximide; CDKi, CDK4/6 inhibitor. (**B**) Cell size was first enlarged by treating cells with 100 nM CDKi for 3 days; then, cells were washed and subjected to chronic HS at 40 °C for 3 more days (CDKi >HS). Cell size (% FSC-MFI; grey part of the bars) and cell death (% annexin V and PI-positive; red part of the bars) were measured by flow cytometry. The values for cell size and death in the different experimental conditions are normalized to the respective 37 °C controls (n=3 biologically independent experiments). (**C**) Cells were first pretreated with 7.5 nM rapamycin (Rapa) for 3 days to reduce the cell size. After that, the cells were subjected to chronic HS at 40 °C for 3 days (Rapa >HS). HS >Rapa, the two treatments were done the other way around. The cell size (% FSC-MFI) and relative cell death (% annexin V and PI-positive) were quantified by flow cytometry. The values for cell size and death in different experimental conditions are normalized to the respective 37 °C control (n=3 biologically independent experiments). (**D**) Scheme of experiments aimed at determining impact of limiting physical space on cell size increase. (**E and F**) Phase-contrast micrographs of RPE1 cells seeded in different numbers to

*Figure 8 continued on next page*

*Figure 8 continued*

restrict the space for cell size increase during adaptation to chronic HS (representative images of n=4 biologically independent experiments). The size bar in the top left panel indicates 100 μM. The cell size (% FSC-MFI) and relative cell death (% annexin V-PI positive) are quantified by flow cytometry. For the bar graphs, the values for cell size and death in different conditions are normalized to the low density (dns) cell population at 37 °C day 0 (n=4 biologically independent experiments). The data are represented as mean values ± SEM for all bar graphs. The statistical significance between the groups was analyzed by two-tailed unpaired Student's t-tests.

The online version of this article includes the following source data and figure supplement(s) for figure 8:

**Source data 1.** Values related to all graphs.

**Figure supplement 1.** Smaller cells are more susceptible to additional stress.

**Figure supplement 1—source data 1.** Values related to all graphs of *Figure 8—figure supplement 1*.

In contrast, we observed that comparatively smaller rapamycin-treated Hsp90α/β KO cells were less prone to apoptosis than the non-treated bigger cells. It remains to be seen whether the ratio of total protein to cell size is slightly more favorable in Hsp90α/β KO cells under these conditions. To further support the importance of mTOR, we treated the larger cells adapted to chronic HS with rapamycin. We observed that WT cells became more apoptotic when we added rapamycin after three days of stress adaptation (*Figure 8C*, *Figure 8—figure supplement 1D*). As rapamycin pre-treatment failed to restrict the cell size increase caused by chronic stress, we then tried to limit the cell size increase with serum starvation (*Figure 8—figure supplement 1E*). We observed that even serum-starved cells enlarged their size in additional chronic stress as they did with rapamycin. Here again, the size enlargement was not up to the level of non-starved cells. We observed that the serum-starved smaller WT cells died more under stress than the non-starved bigger cells (*Figure 8—figure supplement 1E*). As with the rapamycin treatment, in most cases, the serum-starved smaller Hsp90α/β KO cells survived better than the bigger non-starved cells. In both experiments, we did not succeed in preventing the cell size increase upon exposing cells to chronic HS. As a control experiment, we subjected smaller cells obtained with cycloheximide or rapamycin treatment to an acute stress with arsenite for one day. We found that smaller cells were more sensitive to acute stress irrespective of Hsp90 isoform and levels (*Figure 8—figure supplement 1F*). The above-mentioned control experiments also argue that stress resistance is not per se afforded by increased autophagy, as induced by rapamycin, nor by cellular quiescence as induced, for example, by starvation.

We then wondered what would happen if we limited the cell size increase by limiting the available space (*Figure 8D*). We used RPE1 cells, as these cells always grow in single layers and do not grow on top of each other once they are entirely confluent. We observed that at 37 °C, once they become confluent, they do not stop dividing, but become smaller and smaller (*Figure 8E and F*). In contrast, in chronic HS, over time, they enlarge their size and keep dividing when seeded at low density. Next, we seeded a comparatively higher number of cells, such that the culture plate would already be confluent from the beginning, and put it in chronic HS. We observed that the cells in the 'already-confluent' plate at 37 °C kept dividing and shrinking, without obvious cell death (*Figure 8E and F*). However, in chronic HS, the cells in the 'already-confluent' plate, while they were initially able to enlarge their size to some extent, when their size eventually shrank because of the limited space, they started dying off (*Figure 8F*).

## Adaptation to chronic stress requires cytosolic Hsp90 above a threshold level irrespective of Hsp90 isoform

So far, our experiments did not reveal any obvious functional differences between the two cytosolic Hsp90 isoforms. This supports the conclusion that the observed phenotypes in chronic stress, rather than being linked to a specific isoform, are due to below threshold levels of total Hsp90. To address this more directly, we exogenously overexpressed Hsp90α or Hsp90β as mCherry or EGFP fusion proteins, respectively, in Hsp90α/β KO cells. This caused an overall increase in total Hsp90 levels (*Figure 9*). The transfected cells were subjected to chronic HS. After four days, we checked the scaling of total protein by loading an equal volume of cell lysate on a protein gel (see Materials and Methods for details). We found that when the total levels of Hsp90 are elevated by either one of the two isoforms, Hsp90α/β KO cells are able to scale total protein under chronic HS (*Figure 9*). This result complements our

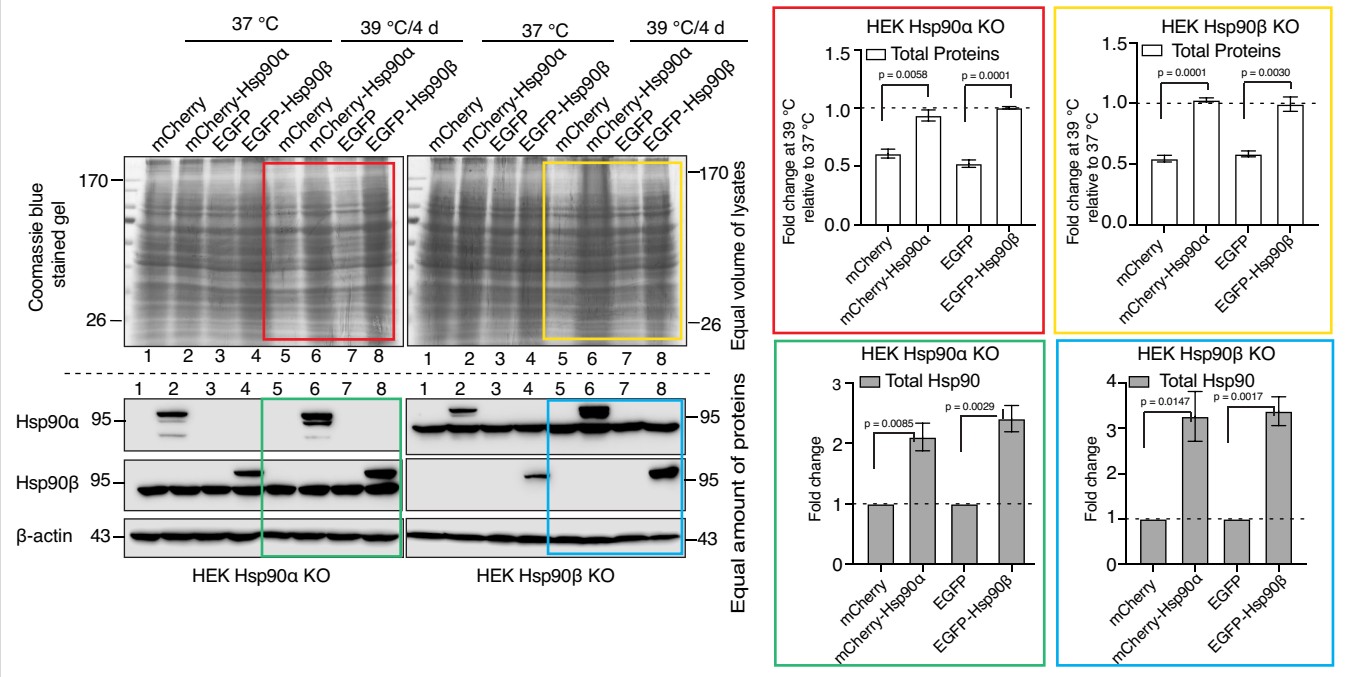

**Figure 9.** Adaptation to chronic stress requires cytosolic Hsp90 above a threshold level. Immunoblots in the lower panels show the endogenous Hsp90α and Hsp90β, and the exogenously overexpressed larger fusion proteins of Hsp90α (as mCherry-Hsp90α) and Hsp90β (as EGFP-Hsp90β). Images of the Coomassie-stained gels in the upper panels show the corresponding levels of total proteins. Colored boxes indicate lanes for samples from cells subjected to chronic HS. The bar graphs on the right show the corresponding quantitation of three biologically independent experiments, with the colored rectangles using the same color code as the rectangles over the immunoblots on the left. The band densities of all blots were normalized to the loading controls. The data are represented as mean values ± SEM. The statistical significance between the groups was analyzed by two-tailed unpaired Student's t-tests.

The online version of this article includes the following source data for figure 9:

**Source data 1.** Raw and annotated immunoblots.

**Source data 2.** Values related to all graphs.

previous findings that Hsp90α/β KO cells become more resistant to both acute and chronic HS upon increasing total Hsp90 levels, irrespective of Hsp90 isoform (*Bhattacharya et al., 2022*).

## Prolonged mild stress leads to excessively bigger cells with an unbalanced cell size to protein ratio associated with senescence

Increased cell size is a hallmark of senescence and aging (*Childs et al., 2015*; *McHugh and Gil, 2018*; *Calcinotto et al., 2019*), whereas we demonstrate here that enlarging cell size is beneficial, at least within the timeframe of our chronic stress paradigm (4–7 days). To determine whether stress-induced cell size enlargement eventually leads to senescence and aging, we maintained RPE1 cells at 39 °C for up to 4 weeks. We observed that these cells continued to enlarge their size (*Figure 10A*). Preliminary results (not shown) suggest this is also true for HEK cells. The cell cycle analysis of RPE1 cells revealed that over time more and more cells became arrested in G1 (*Figure 10B and C*), potentially indicating senescence. To obtain more direct evidence for senescence, we assessed the activity of the senescence-associated marker β-galactosidase (SA-βgal) using flow cytometry. We observed that with chronic stress prolonged beyond two weeks, cells showed a strong increase in SA-βgal activity (*Figure 10D and E*). Intriguingly, when we measured both cell size and total protein over a time course of 4 weeks, we found that cells failed to scale total protein beyond two weeks of mild stress (*Figure 10F*), akin to what had been reported by Neurohr and colleagues for excessively large cells (*Neurohr et al., 2019*). This suggests that the benefits of a larger cell size are lost with much prolonged stress as cells get excessively larger, cannot adjust total protein anymore, and therefore begin to senesce. Thus, while chronic stress initially leads to a protective cell size increase, there is

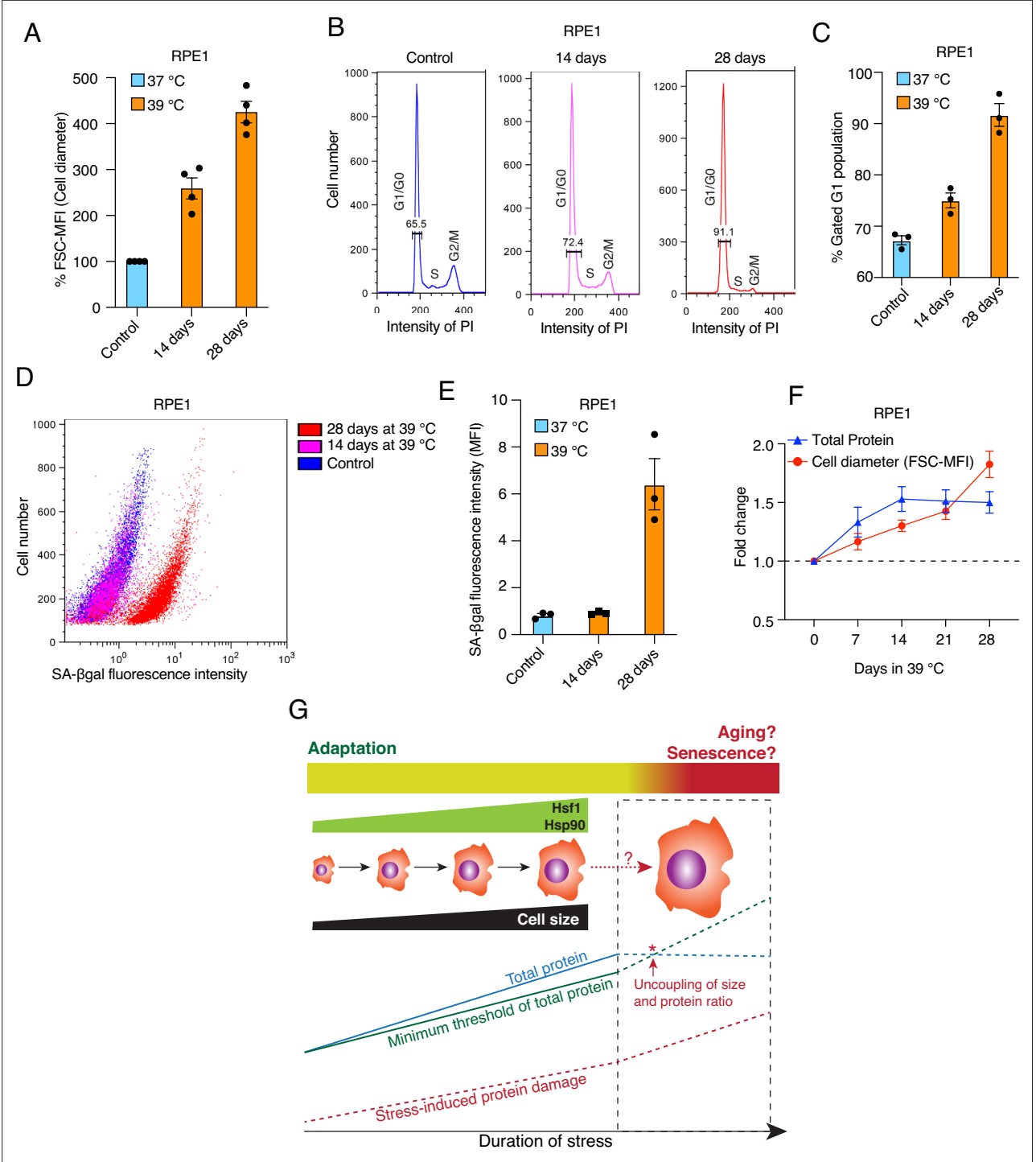

**Figure 10.** Prolonged mild chronic stress triggers excessively bigger cell size and senescence. (**A**) Flow cytometric quantification of cell size of RPE1 cells exposed to prolonged mild HS (n = 4 biologically independent samples). (**B**) Histograms representing the cell cycle distribution of RPE1 cells exposed to prolonged mild HS as indicated, as determined by flow cytometry. (**C**) Quantitation of flow cytometric analyses of the cell cycle (n=3 biologically independent samples). (**D**) Dot plot representing numbers of cells as a function of SA-βgal staining analyzed by flow cytometry. (**E**) Bar graph showing the mean fluorescent intensities of three biologically independent SA-βgal staining experiments of the type shown in panel D. (**F**) Fold change of cell size (represented by the FSC-MFI values) and total proteins (determined as MFI-FL1 values) in prolonged chronic HS, as analyzed by flow cytometry. Cells were fixed, and total proteins were stained using Alexa Fluor 488 NHS ester (n=3 biologically independent experiments). Lines connecting the data points are solely drawn as a visual aid. Note that the quantitative differences of cell size increases between panels A and F may be due to technical differences (live cell versus fixed cell analyses, respectively; see Materials and Methods for further details). (**G**) Schematic representation of the impact

*Figure 10 continued on next page*

*Figure 10 continued*

of chronic mild stress on cells. Wild-type cells initially adapt by enlarging their size and increasing total protein to maintain a minimum threshold level of functional proteins. The right part of the scheme (surrounded by a stippled box), shows what happens if stress persists for much longer: cell size enlargement and total amount of proteins are uncoupled, and because of protein damage, which continues to accumulate, cells become senescent and/or die. The data for all the bar graphs are represented as mean values ± SEM.

The online version of this article includes the following source data for figure 10:

**Source data 1.** Values related to all graphs.

a threshold of chronic stress, possibly in a cell type-specific fashion, beyond which cells cannot cope anymore (*Figure 10G*).

## Discussion

It is now increasingly recognized that exposure to mild environmental stressors is not necessarily detrimental to the organism. Instead, such experiences may foster a resistant or adapted phenotype through hormesis (*Agathokleous and Calabrese, 2022*). Hormesis is a core mechanism of developmental plasticity by which an organism's response to a stressor varies with exposure (*Schirrmacher, 2021*). In contrast, hormetic priming is confined to specific temporal windows and can be considered a preparation to cope with stress. The biological process of aging is claimed to be associated with hormesis (*Gems and Partridge, 2008*). Therefore, physiological cellular responses to mild rather than severe stresses may be more important to address in order to comprehend the biology of aging.

### Cell size enlargement is a prerequisite for chronic stress adaptation

We report that cells adapt to chronic stress by gradually enlarging their size, through a process coupled with increased translation. Stress-induced cell size enlargement is a step towards stress adaptation as part of an intrinsic stress response. Even when we used inhibitors known for restricting the cell size, chronic stress still induced a cell size increase (*Figure 8*, *Figure 8—figure supplement 1*). This suggests that cell size enlargement is a necessity for cells to adapt to prolonged stress. Cell size enlargement has also been linked to cellular senescence, which is a state of long-term cell cycle arrest (*Childs et al., 2015*; *McHugh and Gil, 2018*; *Calcinotto et al., 2019*). In contrast, stress-adapted bigger cells continue to proliferate. It has been reported that bigger cells, when they cannot scale their cellular macromolecules, suffer from cytoplasmic dilution and become senescent (*Neurohr et al., 2019*). The adaptive cell size enlargement in response to chronic stress is coupled with increased translation, which enables cells to maintain their cytoplasmic density and macromolecular crowding. Maintaining macromolecular crowding is necessary to control the kinetics of cellular reactions and to avoid an aging-related deterioration of cellular biochemical processes (*Mourão et al., 2014*). Hence, the adaptive cell size increase in response to chronic stress is different from that associated with cellular senescence. Our observations demonstrate that the coupling of cell size enlargement and translation is essential for cells to adapt to chronic stress. Failure to do so causes their elimination during stress exposure, as we observed with Hsp90-deficient cells.

### Hsp90 enables the rewiring of the stress response

Hsp90-deficient cells subjected to chronic stress suffer from cytoplasmic dilution. For long-term survival under intrinsic or environmentally imposed chronic stress, cells must maintain an equilibrium of protein synthesis, maintenance, and degradation (*Labbadia and Morimoto, 2015*; *Hipp et al., 2019*). Our results indicate that Hsp90 is a key molecule for sustaining translation under these conditions. Unlike during the ISR, in which global cap-dependent protein translation is specifically reduced by the hyperphosphorylation of eIF2α (*Costa-Mattioli and Walter, 2020*; *Persson et al., 2020*), during adaptation to chronic stress, we observed a different phenomenon, which we term "rewiring stress response" (RSR). During RSR, cells initially reduce global translation as happens during an ISR. Then, they turn global translation back on, even when they continue to be exposed to the same stress. This leg of the response requires a threshold level of Hsp90. A hallmark of these transitions is the phosphorylation status of eIF2α and its accompanying impact on global translation. At the beginning of a chronic stress, eIF2α is hyperphosphorylated and global translation is inhibited, whereas prolonged exposure to chronic stress results in the reduction of eIF2α phosphorylation and resumption of

translation (*Figure 6C and E*, *Figure 6—figure supplements 1 and 2*). Thus, the dephosphorylation of eIF2α is central to this phase of the RSR, as it is to terminating the ISR (*Pakos-Zebrucka et al., 2016*). While WT cells switch translation back on, the status of eIF2α phosphorylation and puromycin labeling show that Hsp90α/β KO cells are unable to support this transition, even though they continue to express stress-inducible chaperones and co-chaperones at high levels (*Figure 4*). It appears that Hsp90-deficient cells are stuck at the ISR stage of the RSR. While the ISR is a powerful survival strategy in acute stress, it fails to support the survival of Hsp90-deficient cells during chronic stress adaptation. The ability to mount a RSR is pivotal for the adaptation of cells to chronic stress. The translational recovery and the expression of some of the stress-inducible genes in response to acute stress of the ER and oxidative stress depend on protein phosphatase 1 (PP1) and its regulator GADD34 (*Novoa et al., 2003*; *Carrara et al., 2017*; *Krzyzosiak et al., 2018*). However, our proteomic data indicated that there are no significant changes of PP1 or GADD34 protein levels when one compares cells of different genotypes and stress conditions. Hence, what factors and mechanisms allow cells to transition to RSR remains to be discovered. We report here that a threshold level of total Hsp90 is important to turn on the RSR, not the presence of a particular Hsp90 isoform. If one experimentally elevates the levels of either one of the isoforms, cells can activate the RSR (*Figure 9*).

## Larger cells may be more stress-resistant because of above-threshold levels of macromolecules

When acute stress triggers the ISR, cells only translate selected proteins. With persistent intense stress, the ISR cannot support proteostasis and shifts cells toward apoptosis (*Tian et al., 2021*). When stress is chronic, cells need more than this limited set of proteins. This is where the RSR comes in, allowing cells to get bigger, and to accumulate more protein molecules, and potentially other macromolecules, per cell. It is conceivable that there is an evolutionarily optimized range or a minimal threshold level for all macromolecules, and that this depends on the specific biological needs of a given cell type subjected to a particular type of stress (*Figure 10G*). If the absolute numbers per cell of certain molecules are more important than concentration to adapt and to survive under chronic stress, then bigger cells, which have larger stocks of those molecules, may be better at coping with limited damage than smaller cells. This may even apply to proteins involved in translation, including ribosomal proteins, which did not fully scale even in WT cells. As long as individual cells are above the threshold levels for all relevant macromolecules, they can adapt. Hence, growing bigger is an adaptive response that confers higher stress resistance.

This notion is also supported by our recent findings with mouse KOs (*Bhattacharya et al., 2022*). When we reduced the number of alleles for the cytosolic Hsp90 isoforms from four to one, only embryos that could reestablish the threshold levels of Hsp90 protein by translational reprogramming survived and displayed no apparent phenotype, despite the fact that Hsp90 levels were still lower than those of WT littermates (*Bhattacharya et al., 2022*). There is also evidence that cells can maintain favorable macromolecular crowding by reducing the concentration of highly abundant proteins. The mTORC1 pathway modulates the effective diffusion coefficient within the cytoplasm by reducing the number of ribosomes (*Delarue et al., 2018*). This can avoid increasing molecular crowding, which has been shown to hinder the kinetics of biochemical reactions (*Trappe et al., 2001*; *Zhou et al., 2008*; *Miermont et al., 2013*; *Mourão et al., 2014*). Increased macromolecular crowding might affect protein folding, final shape, conformational stability, binding of small molecules, enzymatic activity, protein-protein interactions, protein-nucleic acid interactions, and pathological aggregation (*Kuznetsova et al., 2014*). Hence, cells that are exposed to prolonged proteotoxic stress might try to maintain a threshold of all the required proteins instead of scaling all proteins at a particular concentration, while tuning molecular crowding to maintain optimal cytoplasmic diffusion coefficients.

## Hsf1-dependent cell size enlargement drives the activation of translation in chronic stress

We found that cell size enlargement during chronic stress depends on Hsf1 activity. This is reminiscent of prior evidence that demonstrated that Hsf1 is linked to the regulation of cell size in different tissues in mammals (*Sakamoto et al., 2006*; *Koya et al., 2013*; *Su et al., 2016*; *Obi et al., 2019*). Hsf1 had been reported to maintain cell growth in a noncanonical way by preserving mTORC1 activity and translation by binding and inactivating the kinase JNK, a known inhibitor of mTORC1 activity (*Su*

*et al., 2016*). In contrast, the cell size increase in response to chronic stress appears to be a canonical Hsf1-mediated HSR phenomenon, which is upstream and independent of the scaling of cellular translation. This is supported by several experiments. Hsp90α/β KO cells subjected to chronic stress could increase their size despite reduced translation. Chronic stress still induced an increase of cell size upon inhibition of translation with rapamycin or cycloheximide, or by serum starvation (*Figure 8*, *Figure 8—figure supplement 1*). We conclude from all of these observations that increased translation is not necessary for cells to increase their size under chronic stress. WT cells start enlarging their size as a response to chronic stress and then translation follows to scale total protein levels. Hsp90α/β KO cells also increase their size in response to stress but fail to couple that to translation, which causes cytoplasmic dilution.

In keeping with the motto 'more is better' for increased stress resistance under chronic stress, we found that cells proportionately increase the size of their nuclei. Although the mechanism and potential adaptive advantage of this organellar enlargement remain to be elucidated, it seems unlikely that it reflects an increase in genomic DNA, such as polyploidy. At the scale of the observed increase in nuclear size, we would have noticed it in our flow cytometric analyses, and cell size enlargement was readily reversible once chronic stress subsided (*Figure 1H*). Heat stress is known to induce chromatin rearrangements, and Hsf1-mediated transcription of repeat sequences and regions of constitutive heterochromatin (*Rizzi et al., 2004*; *Sun et al., 2020*; *Vourc'h et al., 2022*). It is therefore imaginable that genome 'unfolding' underlies the enlargement of the nucleus and that it facilitates a global increase in expression.

## Aging as a failure to adapt to chronic stress

Overall, our findings are relevant to understanding aging. It is well established that mammalian cells get bigger as they grow older (*Cristofalo and Kritchevsky, 1969*; *Treton and Courtois, 1981*; *Demidenko and Blagosklonny, 2008*; *Mammoto et al., 2019*). We speculate that cells enlarge their size with aging as a protective adaptation to accumulating intra- and extracellular stressors. As long as cells manage to scale the biosynthesis of macromolecules such as proteins, their increasing size mitigates the impact of aging (*Figure 10G*). Eventually, the accumulation of stressors, and the breakdown of proper scaling of macromolecules, cannot completely prevent the development of features of aging.

Our findings may have implications for rejuvenation therapies aimed at delaying or even reverting cellular aging (*Blagosklonny, 2019*). Treatment with the mTORC1 inhibitor rapamycin has been reported to improve the function of aged cells and to lengthen the life span of laboratory animals (*Hansen et al., 2007*; *Selman et al., 2009*; *Bjedov et al., 2010*). Despite the low doses that are experimentally used to prevent an age-related decline, there are several adverse side effects. Our data indicate that treatment with mTORC1 inhibitors should be personalized based on the cellular stress level. For healthy cells, where translation plays a vital role in chronic stress adaptation, long-term treatment with rapamycin might affect cellular homeostasis and precipitate aging, or even induce apoptosis. For aged cells with a weakening proteostatic system, rapamycin may rejuvenate cells by reducing translation and inducing autophagy. This is supported by our observation that Hsp90α/β KO cells, where proteostasis is compromised, rapamycin treatment helped the cells to survive better under chronic stress (*Figure 8*, *Figure 8—figure supplement 1*) (see also *Bhattacharya et al., 2022*). We conclude that rather than running the risk of inducing premature aging with prolonged rapamycin treatment from a relatively early age, efforts should focus on developing compounds that induce Hsp90 expression and/or activity, which one could expect to delay aging.

## Physiological changes of cell size

Our observations demonstrate that changes of cell size do not have to be the result of physical constraints, but can be a regulated adaptation. There is evidence for changes in cell size that are induced by physiological conditions (*Ginzberg et al., 2015*), and, of course, aging would be one of them. For example, for rat pancreatic β-cells, insulin secretion, metabolic activity, and global protein production rates are positively correlated with cell size (*Bernal-Mizrachi et al., 2001*). In the kidney, epithelial cells modulate their size in response to fluid flow rates (*Boehlke et al., 2010*). Muscle fibers increase their size in an Hsf1-mediated response to heat generated by exercise (*Obi et al., 2019*). Hsf1 null mice cannot increase the size of their skeletal muscle cells (*Koya et al., 2013*). Furthermore,

Hsf1 plays a critical role in the adaptive increase of the size of cardiac cells (*Sakamoto et al., 2006*). In many organs, when cell numbers decrease due to aging, this is compensated by an increase in cell size to maintain the overall functional capacity (*Ginzberg et al., 2015*). Pancreatic β cells increase their size by over 25% during pregnancy in response to increased insulin demand (*Dhawan et al., 2007*). Similarly, the size of the liver increases during pregnancy through hepatocyte hypertrophy (*Milona et al., 2010*), and liver organ and cell size undergo circadian oscillations (*Sinturel et al., 2017*). Hepatocyte size was also found to increase in organisms continuously exposed to toxic environments (*Hall et al., 2012*). These examples support the idea that it is advantageous for a specific cell type to adopt a specific size under particular physiological conditions. Based on our findings with chronic stress, it would be worth investigating whether Hsp90 more generally supports translational scaling in the context of these physiological changes in cell size.

## Limitations of this study

Hsf1 mediates the cell size increase stimulated by chronic stress, but how chronic stress is sensed and relayed to Hsf1 remains to be established. The underlying mechanism may not be exactly the same as the one involved in mediating acute stress, which itself remains poorly understood. Furthermore, how Hsf1 induces the cell size increase is unknown, and the role of Hsp90 in stimulating translation needs to be dissected in more detail. The latter undoubtedly would be very challenging considering that many of the factors involved in translation, in addition to the eIF2α kinases (*Kudlicki et al., 1987*; *Rose et al., 1987*; *Matts and Hurst, 1989*; *Donzé and Picard, 1999*; *Donzé et al., 2001*; *Marcu et al., 2002*; *Ota and Wang, 2012*; *Zhao et al., 2019*), are likely to be Hsp90 clients. Finally, our experiments with prolonged chronic stress, which demonstrate that eventually even wild-type RPE1 cells cannot maintain homeostasis anymore, raise new questions; above all, it will have to determined whether this is generally true for different cell lines and cell types, and for different types of stresses. Moreover, it is conceivable based on these findings with prolonged stress that there may be limits to how much long-term chronic stress, or combinations of stresses, cells and organisms can handle, even at very low stress levels, with potentially far-reaching consequences.

# Materials and methods

## Key resources table

| Reagent type (species) or resource | Designation | Source or reference | Identifiers | Additional information |
|---|---|---|---|---|
| Cell line (*Homo-sapiens*) | HEK293, Human embryonic kidney cells | ATCC | CRL-3216 | corresponding Hsp90α/β KO cell lines were generated |
| Cell line (*Homo-sapiens*) | A549, human lung epithelial carcinoma cells | ATCC | CCL-185 | corresponding Hsp90α/β KO cell lines were generated |
| Cell line (*Homo-sapiens*) | RPE1, human retinal epithelial cells | ATCC | CRL-4000 | |
| Cell line (*Homo-sapiens*) | HCT116, human colon carcinoma Cells | ATCC | CCL-247 | |
| Cell line (*Mus musculus*) | Mouse adult fibroblasts (MAFs) | *Bhattacharya et al., 2022* | | |
| Recombinant DNA reagent | pcDNA-Flag HSF1 wt (plasmid) | a gift from Len Neckers (*Kijima et al., 2018*) | | transient overexpression of WT Hsf1 |
| Recombinant DNA reagent | pcDNA-Flag HSF1 C205 (plasmid) | a gift from Len Neckers (*Kijima et al., 2018*) | | This express the Hsf1 mutant comprising only the N-terminal 205 amino acids |
| Recombinant DNA reagent | pcDNA3.1(+) (plasmid) | Thermo Fisher Scientific | #V79020 | |
| Recombinant DNA reagent | pCherry.90α (plasmid) | *Picard et al., 2006* | | transient overexpression of Hsp90α |
| Recombinant DNA reagent | pEGFP.90β (plasmid) | *Picard et al., 2006* | | transient overexpression of Hsp90β |

*Continued on next page*

*Continued*

| Reagent type (species) or resource | Designation | Source or reference | Identifiers | Additional information |
|---|---|---|---|---|
| Recombinant DNA reagent | pmCherry-C1 (plasmid) | *Picard et al., 2006* | | Control for pCherry.90α |
| Recombinant DNA reagent | pEGFP-C1 (plasmid) | Clontech | #6084–1 | |
| Recombinant DNA reagent | FUW mCherry-GFP-LC3 (plasmid) | a gift from Anne Brunet (*Leeman et al., 2018*) | Addgene #110060 | autophagy reporter plasmid |
| Recombinant DNA reagent | Ub-M-GFP (plasmid) and Ub-R-GFP (plasmid) | a gift from Nico Dantuma (*Dantuma et al., 2000*) | Addgene #11938 and #11939 | |
| Recombinant DNA reagent | HSE (WT)-Luc (plasmid) | a gift from Ueli Schibler (*Reinke et al., 2008*) | | Hsf1 reporter plasmid |
| Recombinant DNA reagent | pEGFP-Q74 (plasmid) | a gift from David Rubinsztein (*Narain et al., 1999*) | Addgene # 40261 | |
| Recombinant DNA reagent | pRL-CMV (plasmid) | Promega | #E2261 | |
| Recombinant DNA reagent | pSpCas9(BB)–2A-Puro (PX459) (plasmid) | a gift from Feng Zhang (*Ran et al., 2013*) | Addgene #48139 | |
| Recombinant DNA reagent | pGL3-CMV.Luc (plasmid) | a gift from Laurent Guillemot (University of Geneva) | | |
| Sequence-based reagent | shHSF1-1 | This paper, from Microsynth | HSF1 shRNA 3UTR forward | 5'-CCG GGC AGG TTG TTC ATA GTC AGA ACT CGA GTT CTG ACT ATG AAC AAC CTG CTT TTT G-3' |
| Sequence-based reagent | shHSF1-1 | This paper, from Microsynth | HSF1 shRNA 3UTR reverse | 5'-AAT TCA AAA AGC AGG TTG TTC ATA GTC AGA ACT CGA GTT CTG ACT ATG AAC AAC CTG C-3' |
| Sequence-based reagent | shHSF1-2 | This paper, from Microsynth | HSF1 shRNA CDS forward | 5'-CCG GCC AGC AAC AGA AAG TCG TCA ACT CGA GTT GAC GAC TTT CTG TTG CTG GTT TTT G-3' |
| Sequence-based reagent | shHSF1-2 | This paper, from Microsynth | HSF1 shRNA CDS reverse | 5'-AAT TCA AAA ACC AGC AAC AGA AAG TCG TCA ACT CGA GTT GAC GAC TTT CTG TTG CTG G-3' |
| Chemical compound | L-azetidine-2-carboxylic acid (AZC) | Sigma-Aldrich | #P8783 | |
| Chemical compound | Tunicamycin | Cell Signalling | #12819 | |
| Chemical compound | Propidium iodide (PI) | Cayman Chemical | #14289–10 | |
| Chemical compound | Annexin V | Biolegend | #640906 | |
| Chemical compound | Phalloidin-Alexa488 | Thermo Fisher Scientific | #A12379 | |
| Chemical compound | Diamidino-2-phenylindole dye (DAPI) | Thermo Fisher Scientific | #62248 | 1:30,000 in PBS, from 1 mg/ml stock solution |
| Chemical compound, drug | Abemaciclib | MedChemExpress | #HY-16297A-5MG | |
| Chemical compound, drug | Rapamycin | Sigma-Aldrich | #553210 | |
| Commercial assay, kit | Dual-Luciferase detection kit | Promega | #E1910 | |
| Commercial assay, kit | Alexa Fluor 488 NHS Ester (succinimidyl ester) | Thermo Fisher Scientific | #A20100 | |
| Commercial assay, kit | Click-iT Plus OPP Alexa Fluor 594 | Thermo Fisher Scientific | #C10457 | |

*Continued on next page*

*Continued*

| Reagent type (species) or resource | Designation | Source or reference | Identifiers | Additional information |
| --- | --- | --- | --- | --- |
| Commercial assay, kit | N-succinyl-Leu-Leu-Val-Tyr-7-amino-4-methyl-coumarin (suc-LLVY-AMC) | Enzo Life Sciences | #BML-P802-0005 | |
| Commercial assay, kit | CellEvent Senescence Green Flow Cytometry Assay Kit | Invitrogen | #C10840 | |
| Antibody | Anti-GAPDH (Mouse monoclonal) | HyTest Ltd. | 5G4 | 1:1000 |
| Antibody | Anti-Hsp25/27 (Mouse monoclonal) | StressMarq | SMC-114 | 1:1000 |
| Antibody | Anti-puromycin (Mouse monoclonal) | Sigma-Aldrich | MABE343 | 1:22000 |
| Antibody | Anti-Lamin B1 (Rabbit polyclonal) | Cell Signaling Technology | 12586 | 1:1000 |
| Antibody | Anti-Hsp40/Hdj1 (Rabbit polyclonal) | Enzo Lifesciences | ADI-SPA-400 | 1:1000 |
| Antibody | Anti-Hsf1 (Rabbit polyclonal) | Enzo Lifesciences | ADI-SPA-901 | 1:1000 |
| Antibody | Anti-Hsp70 (Mouse monoclonal) | StressMarq | SMC-100 | 1:1000 |
| Antibody | Anti-Phospho-eIF2α (Ser51) (Rabbit polyclonal) | Cell Signaling Technology | 3597 | 1:1000 |
| Antibody | Anti-eIF2α (Rabbit polyclonal) | Cell Signaling Technology | 9722 | 1:1000 |
| Antibody | Anti-Hsp90α (9D2) (Rat monoclonal) | Enzo Lifesciences | ADI-SPA-840 | 1:1000 |
| Antibody | Anti-Hsp90β (scFv H90-10) (Mouse monoclonal) | Geneva Antibody Facility | ABCD_A0870 | 1:2000 |
| Antibody | Anti-mTOR (Rabbit polyclonal) | Cell Signaling Technology | 2983 | 1:2000 |
| Antibody | Anti-Phospho-mTOR (Ser2448) (Rabbit polyclonal) | Cell Signaling Technology | 2971 | 1:1000 |
| Antibody | Anti-Phospho-S6 Ribosomal Protein (Ser235/236) (Rabbit polyclonal) | Cell Signaling Technology | 4858 | 1:1000 |
| Antibody | Anti-S6 Ribosomal Protein (Rabbit polyclonal) | GeneTex | GTX130450 | 1:1000 |
| Antibody | Anti-Hsc70 (Mouse monoclonal) | StressMarq | SMC-151 | 1:2000 |

## Cell lines and cell culture

Human embryonic kidney HEK293T cells (ATCC, CRL-3216), A549 human lung epithelial carcinoma cells (ATCC, CCL-185) (as well as the corresponding Hsp90α/β KO cell lines), and RPE1 human retinal epithelial cells (ATCC, CRL-4000), HCT116 human colon carcinoma cells (ATCC, CCL-247) were maintained in Dulbecco's Modified Eagle Media (DMEM) supplemented with GlutaMAX (Thermo Fisher Scientific #31966047), 10% fetal bovine serum (FBS) (PAN-Biotech #P40-37500), and penicillin/streptomycin (100 u/ml) (Thermo Fisher Scientific #15070063) with 5% $CO_2$ in a 37 °C humidified incubator. We have previously established and characterized MAFs (*Bhattacharya et al., 2022*). Experiments related to MAFs were performed with the cells at 12–24 passages. Human Hsp90α (*HSP90AA1*) and Hsp90β (*HSP90AB1*) KO HEK and A549 cells were generated by the CRISPR/Cas9 gene-editing technology, as reported earlier (*Bhattacharya et al., 2020*; *Bhattacharya et al., 2022*). For A549 Hsp90α/β KO cells, Hsp90α KO clone 1 and Hsp90β KO clone 2 were used for all experiments. For transient transfections, cells were initially seeded at $2\times10^5$ per 2 ml in six-well plates and transfected using PEI MAX 40 K (Polysciences Inc # 24765–100) (1:3 DNA to PEI ratio), except for A549 and its

corresponding Hsp90α/β KO cell lines. For the latter, Lipofectamine LTX (Invitrogen #15338030) was used as directed by the manufacturer's protocol. PEI MAX was notably also used for MAFs, as previously reported (*Bhattacharya et al., 2022*). Cell culture media were changed after 6–8 hr of transfection. Additional chronic stress or treatments were applied 24 hr after transfection.

## Plasmids

For the transient overexpression of WT and mutant Hsf1, plasmids pcDNA-Flag HSF1 wt and pcDNA-Flag HSF1 C205, respectively, were used [a gift from Len Neckers (*Kijima et al., 2018*)]. The plasmid pcDNA-Flag HSF1 C205 allows expression of the Hsf1 mutant comprising only the N-terminal 205 amino acids. pcDNA3.1(+) (Thermo Fisher Scientific # V79020) was used as the empty vector control. For transient overexpression of Hsp90α and Hsp90β, plasmids pCherry.90α, and pEGFP.90β were used (*Picard et al., 2006*). pmCherry-C1 (*Picard et al., 2006*) and pEGFP-C1 (Clontech #6084–1) were used as respective vector control. The autophagy reporter plasmid FUW mCherry-GFP-LC3 was a gift from Anne Brunet (Addgene #110060; *Leeman et al., 2018*). Plasmids Ub-M-GFP and Ub-R-GFP were a gift from Nico Dantuma (Addgene #11938 and #11939; *Dantuma et al., 2000*). The Hsf1 reporter plasmid HSE (WT)-Luc, containing 4 copies of an HSE upstream of a minimal promoter, was a gift from Ueli Schibler (*Reinke et al., 2008*). Plasmid pEGFP-Q74 was a gift from David Rubinsztein (Addgene #40261; *Narain et al., 1999*). Plasmid pRL-CMV was from Promega (#E2261), and pSpCas9(BB)–2A-Puro (PX459) a gift from Feng Zhang (Addgene #48139; *Ran et al., 2013*). Plasmid pGL3-CMV.Luc was a gift from Laurent Guillemot (University of Geneva). For knocking down Hsf1, oligonucleotides (see Key Resources Table) were purchased from Microsynth, annealed, and cloned into pLKO.1 (Addgene #10878). The VSV-G envelope expressing plasmid pMD2.G, and the lentiviral packaging plasmid psPAX2 were gifts from Didier Trono.

## Lentiviral particle generation and gene knockdown

$5 \times 10^6$ HEK cells in a 10 cm plate were co-transfected with plasmids pLKO.1shHSF1-1 or pLKO.1shHSF1-2 (5 µg), pMD2.G (1.25 µg), and psPAX.2 (3.75 µg) with PEI MAX 40 K (Polysciences Inc, # 24765–100) (1:3 DNA to PEI ratio). Suspensions of lentiviral particles were collected and added to the medium of WT and Hsp90α/β KO A549 cells to knock down the expression of Hsf1. Lentiviral control particles were similarly generated and used to express a non-targeting shRNA (not known to target any human mRNA) from plasmid pLKO.1. Transduced cells were selected with 4 µg/ml puromycin (Cayman Chemical # 13884) and used as a pool for further experiments. Knockdowns were validated by immunoblot analyses.

## Chronic mild stress models

The seeding density was $0.5–2 \times 10^6$ cells, depending on the cell line, per 15 cm and 10 cm dishes for chronic HS and all other stresses, respectively.

### Heat shock

To induce chronic HS, HEK and its respective Hsp90α/β KO cells, and HCT116 cells were cultured at 39 °C; A549 and its respective Hsp90α/β KO cells, RPE1 cells, and MAFs were cultured at 40 °C. Depending on the aim of the experiment, cells were cultured in HS for different time spans as mentioned in the respective figure legends.

### Hypoxia

Cells were cultured with 1% $O_2$ and 5% $CO_2$ in a 37 °C humidified incubator for a maximum of 4 days. For HEK and its respective Hsp90α/β KO cells, 5 ml fresh medium was added every day to counter the acidification of the culture medium.

### Oxidative stress

To generate chronic mild oxidative stress, 5–10 µM sodium arsenite (Ars) was added to the cell culture medium for 4 days.

### Proteotoxic stress

For chronic mild proteotoxic stress, 5 µM L-azetidine-2-carboxylic acid (AZC) (Sigma #P8783) was added to the culture medium for 4 days.

### ER stress

For chronic mild ER stress, 250 nM tunicamycin (Cell Signalling #12819) was added to the culture medium for 4 days.

## Acute oxidative stress model

To expose cells to acute oxidative stress, $2–5×10^5$ cells were cultured with 25–40 µM sodium arsenite for 1 day.

## Measurement of cell size

Cells were detached with trypsin/EDTA (PAN-Biotech #P10-024100), collected in complete medium, and thoroughly washed in Tris-buffered saline (TBS). Single-cell suspensions were resuspended in 100–200 µl (PBS) containing 2.5 µg/ml propidium iodide (PI) (Cayman Chemical #14289–10), for 10 min at room temperature (RT). Cells were analyzed with FACS Gallios flow cytometer (Beckman Coulter), and data were analyzed with the FlowJo software package. Cell populations were gated for size measurements based on the value of the forward scatter (FSC) (*Figure 1—figure supplement 2*). We made the assumption that trypsinized cells are round in shape, and therefore, that the respective FSC values are proportional to the diameter of the cells. The PI-positive population was excluded from the size measurement analysis. The respective FSC value was converted to a % value. Control experiments showed that initially preparing the cell suspension in medium containing 5% FCS rather than PBS had no impact on the measured values. However, cell size can be changed if cells are in apoptosis. For the experiments where apoptosis could be observed, cells were stained with PI and annexin V (Biolegend #640906) as described below. Cells positive for PI and annexin V were excluded from the size measurement analysis. Again, controls showed that the annexin V buffer had no impact. For all flow cytometric analyses mentioned here or below, a minimum of 10,000 cells were analyzed for each sample. Additionally, cell diameter was also determined by image analysis using a Roche Innovatis Cedex XS cell counter set to auto-calculation, and for some experiments the area of both nuclei and cells from fluorescence microscopy images using the software ImageJ.

## Cell death assays

### PI staining

Cells were stained with PI as mentioned above and then analyzed by flow cytometry.

### Annexin V-FITC staining

Following a specific treatment, cells were harvested by trypsinization, washed in phosphate-buffered saline (PBS), and resuspended in 100 µl annexin V-binding buffer (10 mM HEPES pH 7.4, 150 mM NaCl, 2.5 mM CaCl$_2$). 5 µl annexin V-FITC and PI to 2.5 µg /ml were added to the cells and incubated for 15 min at 4 °C.

## Cell cycle analyses

Cells were harvested as detailed above. Next, cells were fixed with 70% ice-cold ethanol, washed in PBS, and treated with 100 µg/ml RNase A at RT for 5 min, then incubated with PI to 50 µg/ml for 15–20 min at RT before flow cytometric analysis. Apoptotic cells were identified by the quantitation of the SubG0 (<2 n DNA) cell population (*Figure 1—figure supplement 2*).

## Cell proliferation assays

### Crystal violet staining

To determine the rate of cell proliferation under chronic HS, $5×10^5$ cells/well were seeded into six-well plates and cultured in chronic HS condition as mentioned earlier or at 37 °C. For the heat-adapted cells, they were cultured under chronic HS for 1 week before this assay. One plate was harvested as a control before chronic HS treatment (Day 1). Plates were harvested after 2, 3, and 4 days from

seeding. The proliferation of cells was quantitated by crystal violet staining. Briefly, cells were washed with PBS, incubated with 4% formaldehyde in PBS for 20 min, washed again with PBS, then incubated with 0.1% crystal violet solution in distilled water for 30 min. The wells were washed thoroughly with distilled water and air-dried overnight. Crystals were dissolved in glacial acetic acid, and the absorbance was measured at 595 nm with a Cytation 3 Image Reader (Agilent).

## Cell counting

To determine the impact of chronic HS on cell proliferation, HEK and A549 cells were seeded at a density of $3–5×10^6$ cells per 20 ml in a 15 cm plate and subjected to mild HS at 39 °C and 40 °C for HEK and A549 cells, respectively. A parallel set was maintained at 37 °C as control. Every 7 days, cells were harvested by trypsinization and counted using a hemocytometer under the light microscope with the trypan blue exclusion assay.

## Phase contrast and fluorescence microscopy

Cellular morphology and cell culture density were analyzed using an inverted light microscope (Olympus CK2), and phase contrast images were captured with a Dino-lite camera using the software DinoXcope. To visualize the impact of chronic HS on the actin network, cells were grown under chronic HS conditions on a poly-L-lysine-coated coverslip for 4 days, before being washed with PBS and fixed with 3% glyoxal solution (*Richter et al., 2018*). Actin was stained with Phalloidin-Alexa488 (Thermo Fisher Scientific #A12379), and the nucleus was stained with diamidino-2-phenylindole dye (DAPI) (1:30,000 in PBS, from 1 mg/ml stock solution (Thermo Fisher Scientific #62248)). Coverslips were mounted on glass slides using Mowiol. Cells were visualized, and images were captured with a fluorescence microscope (Zeiss, Germany).

## Hsf1 activity assay

To check the Hsf1 activity at 37 °C, under chronic HS, and upon capsaicin treatment, at first, $4×10^4$ cells per 400 μl were seeded in 24-well plates. Cells were co-transfected with the Hsf1 luciferase reporter plasmid HSE (WT)-Luc and the Renilla luciferase internal transfection control (plasmid pRL-CMV). 24 h after transfection, cells were cultured under chronic HS conditions for 2 days or in capsaicin (10–250 μM) for 4 days. Then, cells were lysed, and firefly and Renilla luciferase activities were measured using the Dual-Luciferase detection kit (Promega #E1910) with a bioluminescence plate reader (Citation, BioTek). Firefly luciferase activities were normalized to those of Renilla luciferase. The corresponding fold change of Hsf1 activity under chronic HS was normalized to their respective Hsf1 activity at 37 °C. In experiments where Hsf1 activity was measured after overexpressing WT or mutant Hsf1, HSE (WT)-Luc and pRL-CMV were co-transfected with the respective expression plasmids.

## FRAP experiments

FRAP experiments were carried out on a Leica SP8 confocal microscope using a 63 x oil immersion objective. For in vivo measurements of cytoplasmic density, cells were transiently transfected with EGFP expression plasmid pEGFP-C1. Fluorescence was bleached within a circular area of 10 μm$^2$ with homogeneous fluorescence within the cytoplasm and a background zone outside the cell. After bleaching, images were taken every 1.29 s for HEK and its corresponding Hsp90α/β KO cells and 0.22 s for A549 and its corresponding Hsp90α/β KO cells for a total of 30 images. The data were used for the calculation of t-half and diffusion coefficients by a one-phase exponential association function, and recovery curves were built using GraphPad Prism 8 (GraphPad Software, Inc, La Jolla, CA).

## Protein extraction and analysis

### Quantification of cellular protein

To quantify the amount of total soluble protein per cell, $5x10^6$ cells per experimental condition were harvested, pellets were washed at least twice with PBS, and then lysed in ice-cold lysis buffer (20 mM Tris-HCl pH 7.4, 2 mM EDTA, 150 mM NaCl, 1.2% sodium deoxycholate, 1.2% Triton X-100, protease inhibitor cocktail (Thermo Fisher Scientific #78429), and phosphatase inhibitor cocktail (Thermo Fisher Scientific #78420)). Cell lysates were sonicated for 15 min at high power with the Bioruptor sonicator (Diagenode). Protein quantification was performed using the Bradford reagent (Biorad #5000001), measuring absorbance at 595 nm. To visualize the amount of total protein in each experimental

condition, the same volume of cell lysates was mixed with SDS sample buffer and run on 10% SDS-PAGE, and proteins were stained with Coomassie blue for 1 hr and de-stained overnight in water.

## Analysis of cell size to total protein ratio

Cells from different experimental conditions were fixed for 10 min in 4% formaldehyde, washed with PBS, and permeabilized in 100% methanol for 10 min at –20 °C. Methanol was removed, and cells were washed once with 0.2 M sodium bicarbonate, followed by staining in 0.5 ml 0.2 M sodium bicarbonate containing 50 mg/ml Alexa Fluor 488 NHS Ester (succinimidyl ester) (Thermo Fisher Scientific #A20100) for 30 min at RT. Cells were then analyzed using flow cytometry. The amount of protein per cell was quantitated as log of the mean fluorescent intensity (MFI) values (detected in the FL1 channel, i.e. MFI-FL1). Relative cell size was deduced from the FSC values (see above).

## Assay of protein translation

To measure de novo protein translation, cells were seeded at a density of $1.2–1.5×10^6$ per 10 ml in a 10 cm plate. Cells were then treated with 1 µM puromycin for 0–2.5 hr, while being cultured at 37 °C or 39 °C. Cells were harvested and lysed in lysis buffer (20 mM Tris-HCl pH 7.4, 2 mM EDTA, 150 mM NaCl, 1.2% sodium deoxycholate, 1.2% Triton-X100, 200 mM iodoacetamide, protease inhibitor cocktail). A total of 75 µg of clarified cell lysates were separated by 10% SDS-PAGE and immunoblotted for newly synthesized proteins or polypeptides with anti-puromycin antibodies. The intensities of the corresponding lanes reflect the total amount of global protein translation for that time point. To determine de novo protein translation at the single cell level in relation to cell size, cells were maintained at 39 °C for 4 hr, 1 day, and 4 days. Nascent polypeptides were labelled using Click-iT Plus OPP Alexa Fluor 594 (Thermo Fisher Scientific #C10457) as described in the manufacturer's protocol. Briefly, O-propargyl-puromycin (OPP) (2 µM) was added to cultured cells up to 4 hr after the above-mentioned incubation at 39 °C. In parallel, a control set was done at 37 °C. Cells were then washed with PBS, fixed in 3.7% formaldehyde in PBS, permeabilized in 0.5% Triton X-100 in PBS, and incubated at RT for 15 min. The permeabilization buffer was washed off with PBS. Click-iT Plus OPP reaction cocktail was added and incubated for 30 min at RT, protected from light. The reaction cocktail was removed and washed once with 1 ml of Click-iT Reaction Rinse Buffer, further washed with PBS, and kept in PBS until newly synthesized proteins were quantitated using flow cytometry (*Figure 6—figure supplement 3*) and expressed as log MFI values, and relative to cell size.

## Immunoblot analyses

Lysates of cells (20–100 µg) were subjected to SDS-PAGE and transferred onto a nitrocellulose membrane (GVS Life Science) with a wet blot transfer system (VWR #BTV100). Membranes were blocked with 2–5% non-fat dry milk or bovine serum albumin in TBS with 0.2% Tween 20 (TBST) and incubated with primary antibodies overnight at 4 °C. Then they were washed with TBST, incubated with the corresponding HRP-conjugated antibodies for 1 hr at RT, and developed using the WesternBright chemiluminescent substrate (Advansta #K-12045-D50). Images were captured using a LI-COR Odyssey or Amersham ImageQuant 800 image recorder. The EZ-Run Prestained Protein Ladder (Thermo Fisher Scientific) or the AcuteBand Prestained Protein ladder (Lubioscience) were used as protein molecular weight markers. Details of the primary antibodies and antisera are given in the Key Resources Table.

## Autophagic flux measurement

Cells seeded in six-well plate at a density of $4×10^5$ per 2 ml were transfected with the autophagy reporter plasmid FUW mCherry-GFP-LC3 as described above. 24 hr after transfection, cells were subjected to chronic HS for 2 days. Cells were harvested by trypsinization, and GFP- and mCherry-positive cells were measured by flow cytometry (*Figure 7—figure supplement 2*). The autophagic flux was measured by calculating the ratio of the MFI of mCherry and GFP-positive cells. A higher relative ratio of mCherry/GFP is indicative of a higher autophagic flux.

## In vivo UPS activity assay

Cells seeded at a density of $5×105$ per 2 ml in six-well plates were transfected with plasmids Ub-M-GFP and Ub-R-GFP for expression of stable and degradation-prone GFP, respectively (*Dantuma et al.,*

*2000*). The next day, the medium was changed, and cells were cultured at 37 °C or 39 °C for 2 days. Cells were then harvested by trypsinization and GFP-positive cells were quantitated by flow cytometry. The in vivo UPS activity was expressed as the MFI of Ub-M-GFP-positive cells minus the MFI of Ub-R-GFP-positive cells (*Figure 7—figure supplement 2*). Fold change of UPS activity in HS was normalized to the corresponding activity at 37 °C. Fold change of UPS activity of Hsp90α/β KO cells was also normalized to the fold change relative to WT cells.

### In vivo luciferase refolding assay

Cells cultured at 37 °C or cells adapted at 39 °C for 2 days were transfected with the luciferase expression vector pGL3-CMV.Luc. Cells were subjected to an acute HS at 43 °C for 15 min to denature luciferase, followed by incubation at 37 °C for the control cells or at 39 °C for the heat-adapted cell for 1–3 hr to allow refolding of luciferase. Cells were harvested by centrifugation and lysed with the Passive Lysis Buffer of the Dual-Luciferase detection kit (Promega). Ten µl cell extract was mixed with an equal volume of firefly luciferase assay substrate from the kit, and the luciferase luminescence signals were measured with a bioluminescence plate reader (Citation, BioTek).

### In vitro proteasomal activity assay

Cells cultured at 37 °C or at 39 °C were harvested and washed. Cell pellets were resuspended in lysis buffer (25 mM Tris-HCl pH 7.4, 250 mM sucrose, 5 mM $MgCl_2$, 1% IGEPAL, 1 mM DTT, 1 mM ATP) and incubated for 10–15 min on ice. Samples were centrifuged at $16,100 \times g$ for 20 min, and supernatants were collected for the proteasomal activity assay. Equal amounts of protein (50 µg) for each sample were diluted in proteasomal reaction buffer (50 mM Tris-HCl pH 7.4, 5 mM $MgCl_2$, 1 mM DTT, 1 mM ATP) in a 96-well opaque bottom white plate and 50 µM N-succinyl-Leu-Leu-Val-Tyr-7-amino-4-methyl-coumarin (suc-LLVY-AMC) (Enzo Life Sciences, #BML-P802-0005) was added to each well. AMC fluorescence was measured at 460 nm for 5–60 min, with an excitation at 380 nm. All fluorescence measurements were recorded using a plate reader (Cytation 3, BioTek).

### Protein aggregation

To analyze polyglutamine (polyQ) protein aggregation within cells, cells were seeded on glass coverslips and transfected with the plasmid pEGFP-Q74 and 24 hr after transfection, cells were placed at 39 °C and the respective controls at 37 °C. 42 hr after transfection, cells were fixed with 4% paraformaldehyde and mounted on glass slides using Mowiol. EGFP-positive cells were visualized, and images were captured with a fluorescence microscope (Zeiss, Germany).

### Increase and reduction of cell size with various inhibitors

To increase the cell size, $2 \times 10^5$ cells in 3 ml were seeded in six-well plates and treated with the Cdk4/6 inhibitor abemaciclib (MedChemExpress, #HY-16297A-5MG) at 50–100 nM for 2–3 days. At this point, one experimental set was harvested. To reduce the cell size, $2 \times 10^5$ cells in 3 ml were seeded in six-well plates, and treated with 5–10 nM rapamycin (Sigma-Aldrich, #553210), or 100 ng/ml cycloheximide (CHX) for 2–3 days. Alternatively, cell size was reduced by replacing the standard growth medium 24 hr after seeding with medium containing only 1% FBS. Parallel sets that were not harvested were washed with fresh medium and subjected to other stress treatments. Cell size and cell death were determined by flow cytometry as described above.

### SA-βgal activity assays

Cells cultured at 37 °C or at 39 °C were harvested and washed. SA-βgal activity was measured using a CellEvent Senescence Green Flow Cytometry Assay Kit (Invitrogen, #C10840) as described by the manufacturer's protocol.

### Proteomic analyses
#### Protein digestion
HEK WT and their respective Hsp90α/β KO cells were cultured at 37 °C, and at 39 °C for 1 day and 4 days. Cells were harvested and snap-frozen. Replicate samples were digested according to a modified version of the iST method (named miST method) (*Kulak et al., 2014*). Briefly, frozen cell pellets were resuspended in 3 ml miST lysis buffer (1% sodium deoxycholate, 100 mM Tris pH 8.6, 10 mM

DTT). Resuspended samples were sonicated and heated at 95 °C for 5 min. After quantification with tryptophan fluorescence, 100 µg of samples in 50 µl buffer were diluted 1:1 (v:v) with water. Reduced disulfides were alkylated by adding ¼ volume of 160 mM chloroacetamide (final 32 mM) and incubating at 25 °C for 45 min in the dark. Samples were adjusted to 3 mM EDTA and digested with 1.0 µg Trypsin/LysC mix (Promega #V5073) for 1 hr at 37 °C, followed by a second 1 hr digestion with a second aliquot of 0.5 µg trypsin. To remove sodium deoxycholate, two sample volumes of isopropanol containing 1% trifluoroacetic acid (TFA) were added to the digests, and the samples were desalted on a strong cation exchange (SCX) plate (Oasis MCX; Waters Corp., Milford, MA) by centrifugation. After washing with isopropanol/1% TFA, peptides were eluted in 200 µl of 80% acetonitrile, 19% water, 1% (v/v) $NH_3$.

## Peptide fractionation for library construction

After redissolution of samples in 1.0 ml of loading buffer (2% acetonitrile with 0.05% TFA), aliquots (10 µl) of samples were pooled and separated into 6 fractions by off-line basic reversed-phase (bRP) using the Pierce High pH Reversed-Phase Peptide Fractionation Kit (Thermo Fisher Scientific). The fractions were collected in 7.5, 10, 12.5, 15, 17.5, and 50% acetonitrile in 0.1% triethylamine (~pH 10). Dried bRP fractions were redissolved in 40 µl loading buffer, and 4 µl were injected for LC-MS/MS analyses.

## LC-MS/MS

LC-MS/MS analyses were carried out on a TIMS-TOF Pro (Bruker, Bremen, Germany) mass spectrometer interfaced through a nanospray ion source ("captive spray") to an Ultimate 3000 RSLCnano HPLC system (Dionex). Peptides were separated on a reversed-phase custom-packed 40 cm C18 column (75 µm ID, 100 Å, Reprosil Pur 1.9 µm particles, Dr. Maisch, Germany) at a flow rate of 0.250 µl / min with a 2–27% acetonitrile gradient in 93 min followed by a ramp to 45% in 15 min and to 90% in 5 min (all solvents contained 0.1% formic acid). Identical LC gradients were used for DDA and DIA measurements.

## Library creation

Raw Bruker MS data were processed directly with Spectronaut 15.4 (Biognosys, Schlieren, Switzerland). A library was constructed from the DDA bRP fraction data by searching the reference human proteome (https://www.uniprot.org/; accessed on September 3rd, 2020, containing 75,796 sequences). For identification, peptides of 7–52 AA length were considered, cleaved with trypsin/P specificity, and a maximum of 2 missed cleavages. Carbamidomethylation of cysteine (fixed), methionine oxidation and N-terminal protein acetylation (variable) were the modifications applied. Mass calibration was dynamic and based on a first database search. The Pulsar engine was used for peptide identification. Protein inference was performed with the IDPicker algorithm. Spectra, peptide, and protein identifications were all filtered at 1% FDR against a decoy database. Specific filtering for library construction removed fragments corresponding to less than 3 AA and fragments outside the 300–1800 m/z range. Also, only fragments with a minimum base peak intensity of 5% were kept. Precursors with less than three fragments were also eliminated, and only the best six fragments were kept per precursor. No filtering was done on the basis of charge state, and a maximum of 2 missed cleavages was allowed. Shared (non-proteotypic) peptides were kept. The library created contained 119,573 precursors mapping to 91,154 stripped sequences, of which 35,119 were proteotypic. These corresponded to 8,632 protein groups (12,252 proteins). Of these, 959 were single hits (one peptide precursor). In total 701,091 fragments were used for quantitation.

## DIA quantitation

Peptide-centric analysis of DIA data was done with Spectronaut 15.4 using the library described above. Single hits proteins (defined as matched by one stripped sequence only) were kept in the Spectronaut analysis. Peptide quantitation was based on XIC area, for which a minimum of 1 and a maximum of 3 (the 3 best) precursors were considered for each peptide, from which the median value was selected. Quantities for protein groups were derived from inter-run peptide ratios based on MaxLFQ algorithm (*Cox et al., 2014*). Global normalization of runs/samples was done based on the median of peptides.

## Data processing and statistical tests

All subsequent analyses were done with the Perseus software package (version 1.6.15.0). Intensity values were log2-transformed, and after assignment to groups, t-tests were carried out with permutation-based FDR correction for multiple testing (q-value threshold <0.01). The difference of means obtained from the tests was used for 1D enrichment analysis on associated GO/KEGG annotations as described (*Cox and Mann, 2012*). The enrichment analysis was also FDR-filtered (Benjamini-Hochberg, q-value <0.02). The data of the proteomic analysis are supplied as *Source data 1*.

## Pathway enrichment analysis and visualization

The ordered lists of differentially expressed genes obtained for each pair of comparison were used to do GSEA (*Subramanian et al., 2005*) on GO term databases with functions of the R package clusterProfiler (version 3.18.1; *Yu et al., 2012*). Pathways significantly enriched (q-value <0.05) were visualized with ggplot2 scripts. Redundant pathways and pathways irrelevant for the cell type were removed.

## Ribosome and polysome fractionation

Polysome fractionation was performed at the 'BioCode: RNA to Proteins' core facility of the Faculty of Medicine of the University of Geneva. Cells were cultured at 37 °C or 39 °C for 4 days. Before harvesting the cells, 100 µg/ml CHX was added to the medium, and cells were maintained in the respective incubator for 15–20 min. Cells were then harvested and washed in ice-cold PBS containing 100 µg/ml CHX, snap-frozen and proceeded to the next steps. Cells were weighed, and resuspended in lysis buffer (50 mM Tris-HCl pH 7.4, 100 mM KCl, 1.5 mM $MgCl_2$, 1.5% Triton X-100, 1 mM DTT, 100 µg/ml CHX, 1 mg/ml heparin, 25 u/ml Turbo DNase I (Roche, #04716728001), 25 u/µl SUPERaseIn RNase inhibitor (Invitrogen, #AM2694), protease inhibitors (Roche, #04693132001)) at 200 µl lysis buffer per 100 µg pellet. Cell suspensions were passed through a 25 G needle 10–12 times. Cell debris were pelleted at 20,000 *g* at 4 °C for 20 min. The total RNA concentration of the supernatant was determined. Cell lysates containing 350 µg of total RNA were loaded onto linear 20–60% sucrose gradients prepared with the gradient buffer (50 mM Tris-HCl pH 7.4, 100 mM KCl, 1.5 mM $MgCl_2$, 1 mM DTT, 100 µg/ml CHX). Ribosomes/polysomes were centrifuged at 247,600 *g* (38,000 rpm) in a SW41 Ti rotor (Beckman Coulter, #331362) for 3 h 30 min at 4 °C. Fractionated ribosomes/polysomes were measured and collected using a density gradient fractionation system (ISCO).

## General data analyses

Data processing and analyses were performed using GraphPad Prism (version 8).

## Acknowledgements

We are indebted to Manfredo Quadroni of the PAF of the University of Lausanne for the proteomic analyses. We are grateful to the Bioimaging Center of the Faculty of Science of the University of Geneva for assistance with the confocal microscopy. This work has been supported by the Swiss National Science Foundation (grant 31003 A_172789/1) and the Canton de Genève.

## Additional information

### Funding

| Funder | Grant reference number | Author |
| --- | --- | --- |
| Schweizerischer Nationalfonds zur Förderung der Wissenschaftlichen Forschung | 31003A_172789/1 | Didier Picard |
| Canton de Genève | | Didier Picard |

| Funder | Grant reference number | Author |
|--------|------------------------|--------|

The funders had no role in study design, data collection and interpretation, or the decision to submit the work for publication.

## Author contributions

Samarpan Maiti, Conceptualization, Data curation, Formal analysis, Investigation, Visualization, Methodology, Writing - original draft, Writing - review and editing; Kaushik Bhattacharya, Conceptualization, Formal analysis, Investigation, Visualization; Diana Wider, Investigation; Dina Hany, Olesya Panasenko, Lilia Bernasconi, Formal analysis, Investigation; Nicolas Hulo, Data curation, Formal analysis; Didier Picard, Conceptualization, Supervision, Funding acquisition, Writing - original draft, Project administration, Writing - review and editing

## Author ORCIDs

Samarpan Maiti ⓘ http://orcid.org/0000-0001-9090-2398
Nicolas Hulo ⓘ http://orcid.org/0000-0003-2640-636X
Didier Picard ⓘ http://orcid.org/0000-0001-8816-9668

Reviewer #1 (Public Review): https://doi.org/10.7554/eLife.88658.3.sa1
Reviewer #2 (Public Review): https://doi.org/10.7554/eLife.88658.3.sa2
Author Response https://doi.org/10.7554/eLife.88658.3.sa3

# Additional files

## Supplementary files

- MDAR checklist
- Source data 1. Proteomics data.

## Data availability

The mass spectrometry proteomics data have been deposited to the ProteomeXchange Consortium via the PRIDE partner repository with the dataset identifier PXD039672, and for a subset, they are available in *Source data 1*. All other data needed to evaluate the conclusions in the paper are presented in the paper. Newly generated reagents are available upon request.

The following dataset was generated:

| Author(s) | Year | Dataset title | Dataset URL | Database and Identifier |
|-----------|------|---------------|-------------|-------------------------|
| Maiti et al | 2023 | Molecular chaperone Hsp90 supports adaptive cell size increase in response to chronic stress | https://proteomecentral.proteomexchange.org/cgi/GetDataset?ID=PXD039672 | ProteomeXchange, PXD039672 |

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
