## [Editor Report · eLife assessment]

This **important** study describes the coordinated regulation of cellular size and protein translation in response to chronic stress as an adaptive mechanism, termed the 'rewiring stress response' regulated by the heat shock response. The evidence supporting this conclusion is **solid**, utilizing diverse methods to monitor and manipulate cell size and evaluate stress resistance. The study could be strengthened by the inclusion of more experiments focused on defining the mechanistic basis of this coordination and broadening the scope of the specific role of the 'rewiring stress response' across different chronic cellular stresses. This work will be of broad interest to researchers interested in diverse fields including cellular proteostasis, stress-responsive signaling, and aging and senescence.

---

## [Referee Report · Reviewer #1 (Public Review)]

The manuscript describes that cultured mammalian cells adapt to chronic stress by increasing their size and protein translation through Hsp90. The authors extensively use Hsp90 knockout cells and mass spectrometry to provide solid evidence that chronic heat shock response is accompanied by cell size changes and stress resistance in large cells. The major strength of the work is the authors ability to document the heat shock response in detail. The increased stress resistance of large cells is conceptually important and provides one potential explanation why cells need to control their size. This work adds to our understanding of how cellular stress is managed, and while stress responses have been observed previously in relation to cell size, this work provides evidence for increased stress resistance in larger cells.

---

## [Referee Report · Reviewer #2 (Public Review)]

The authors have done a number of additional experiments and textual changes to address referee comments from the first round of review that have improved some aspects of the manuscript. However, they did not fully address two major issues brought up in my previous public review, reiterated below.

1. What is the specific role for HSP90a/b in regulating protein translation during chronic stress through the ISR or related pathways? The authors indicate that the induction of the eIF2a phosphatase GADD34 is not impacted in HSP90-deficient cells, so what role does HSP90 have in this process. Is HSP90 required for proper folding of GADD34? Would you see similar effects in protein translation recovery if other ISR activators are used in HSP90-deficient cells?

2. Are similar effects observed in non-dividing cells?' Does chronic stress lead to increases of size and regulation of protein translation in primary cell models that are not undergoing division.

This leaves the study as an interesting observational study that correlates increases in cell size and protein translation. However, it doesn't really answer some of the most important questions related to mechanisms defining this correlation. Regardless, this remains an interesting jumping off point to continue exploring this interesting finding correlating cell size and stress signaling that will be further pursued in subsequent manuscripts, which will likely continue to reveal the importance and mechanistic basis of this 'rewiring stress response' during stress and in disease.

---

## [Author Response]

The following is the authors’ response to the original reviews.

We have substantially revised our manuscript based on the extensive and highly constructive comments of the reviewers. We have included new data, refined existing data, and revised the text. To do this, some figures had to be split and several figures had to be renumbered. The additional experiments presented at the end of the Results also led us to expand our discussion of current limitations of our story.

Recommendations for the authors

**Reviewer #1:**
To improve the manuscript, I have some recommendations for the authors.1. The cell size was quantified using flow cytometry (forward scatter). While this approach provides a convenient way to measure cell size, it is only a relative way to compare the cell size. A 10% increase in FSC value does not necessarily mean a 10% increase in diameter, this depends on the instrument. Consequently, the claims of density changes such as based on the panel 5B may be incorrect. It would be useful also to perform some experiments with Coulter Counter or imaging based quantification of cell size.

We agree and this is precisely why we had also measured cell diameters by imaging (reported at the bottom of page 7 and figure supplement 1D in the initial version of the manuscript). In the revised manuscript, we have added a cautionary note in the same context. Regarding density changes, those measurements by FRAP are independent of assumptions about cell diameter. When cell density is down and cells are larger by whatever factor, one can safely conclude that total protein did not scale.

1. When the Hsp90a/b KOs are introduced on page 9, it would be helpful to know at this stage whether the double KO cells are viable to understand why the individual KOs rather than double knockout cells were used.

We have now added a statement to indicate that total Hsp90 KOs are not viable in eukaryotes.

1. How the following can be reconciled with previous work is a bit unclear and needs some clarification: Neurohr et al 2019 identifies cytoplasmic dilution in larger cells, but in this manuscript WT cells maintain the same cytoplasmic density while becoming larger under chronic stress while the Hsp90 KO cells have reduced cytoplasmic density. Does this mean that the cytoplasmic dilution does not relate to cell size but is indirect and related to heat stress? Or is this related to uncoupling of cell size and density only in excessively large cells as for example HEK cells only increase their diameter by 30% based on the flow cytometry analysis?

Yes, indeed, beyond a certain threshold, excessively enlarged cells cannot scale protein anymore. In the revised manuscript, we not only look at cells exposed to stress for much longer (up to one month) (see last paragraph of revised Results). These cells become even bigger, and in agreement with Neurohr and colleagues, we find that protein scaling breaks down.

1. Related to the previous, the authors state that "Hsp90 levels rather than a specific isoform are critical for maintaining the cytoplasmic density", but there is no direct evidence connecting Hsp90 levels to cell size. Given the number of proteomics experiments done in this work, can a correlation between Hsp90 levels and cell size/cell density be identified? Or is this related to the way cell size is increased in chronic stress as later the authors say that with the CDK4/6 inhibitor Hsp90α/β KO cells can scale the total protein.

We have previously determined total Hsp90 levels quantitatively by mass spec (Bhattacharya et al., 2022; see Figure S8 there) (now explicitly mentioned in the same context as our revision related to point #2, see above), and we have now also added the quantitation, including that of total Hsp90 levels, in what is now Figure 9.

1. Page 17 states "Hsp90α/β KO cells increase cell size while translation is still reduced. Thus, cell size and translation must be coupled for adaptation to chronic stress." This feels like an important conclusion of the paper, yet the direct evidence is rather limited and the authors are clearly not sure how the Hsp90 KO cells increase their size without increasing the translational capacity. Yes, a potential explanation is provided immediately afterward as the authors show that Hsp90α/β KO cells subjected to chronic HS also have reduced proteasomal activity. Reducing protein degradation allows cells to gain more protein even if the synthesis rate does not increase (steady-state protein levels is a balance between synthesis and degradation). As stated by the authors in the discussion, the KO cells "fail to couple cell size increase to translation" simply because they can increase total protein, and cell size, by reducing protein degradation.

Yes, reducing protein turnover might be a viable strategy, but here, reduced protein degradation in the Hsp90 KOs is clearly not enough since total protein levels cannot keep up with the cell size increase.

1. What is unclear to me is to what extent these results (where chronic heat stress increases cell size and cells proliferate) relate to large senescent cells which are arrested. The discussion speculates that a failure to adapt to stress leads to aging, but direct evidence is lacking.

Even though we feel quite strongly that (some) speculation should be allowed, we now provide more direct evidence for senescence (see Figure 10 of revised manuscript and corresponding text). Moreover, we had already demonstrated in Bhattacharya et al. 2022 that senescence is triggered by below-threshold levels of Hsp90 (i.e. cells express senescence markers). But note that senescence is only manifest upon prolonged exposure to chronic mild stress, and that our standard protocol for chronic mild stress was established in such a way as to avoid much of an effect on viability and proliferation (see Figure 1). So no, at least for wild-type cells, except for the experiments of Figure 10, what we studied are not large senescent and arrested cells.

1. The clarity and content of the figures need some improvement. For example, in Fig 1, it is difficult to see the small symbols specifying the cell lines as the replicates are often overlapping. The font for p values is also too small. For Fig 2, legend says "the statistical significance between the groups was analyzed by two-tailed unpaired Student's t-tests." but there are no statistics shown. The use of statistical testing is also inconsistent across different figures and panels, for example Fig 3 A vs 3C and 5A vs 5H. In Fig 4. the legend talks about p-value, but y axis in panels is q value. The authors need to clarify this by mentioning that these are adjusted p values. Fig 7. should also explain "Rapa" in the legend or state "Rapamycin" in the figure.

To avoid overloading figures further with enlarged text, we prefer not to increase the font size of the p-values, and for graphs where data points are too small or overlap, we remind the reviewer that all original data will be available with the paper (and linked to from each figure). For Figure 2, we removed the indicated orphaned statement. We've now added stats for Figure 3C, and double checked all others; note that in most cases where the differences are really obvious, we did not add p-values. Wherever there were q-values as Y axis, we have now also added the term "adjusted p-value" in the legend. As for "Rapa", it was and still is defined in the legend.

1. The data in Fig 5A looks curious as the 39C response is bimodal suggesting that only some cells adapt to the heat stress or could this be a technical issue with the measurements?

The reason for this is that the data points are from 2 independent experiments. This means that the measurements were done on different days with a microscope that had to be calibrated again and may have been in a slightly different mode. This is not uncommon with this type of data. As an example of that, please see Fig. 3C of Persson et al. Cell 183:1572-1585 (https://doi.org/10.1016/j.cell.2020.10.017).

**Reviewer #2:**
Specific comments for authors:Major comments:1. Fig. 1F: if cells are not split for 7 days than they start growing in multi-layers. The density within a plate affects their proliferation rate as well as their translation rate. Therefore, a proliferation curve (with counting) when cells are kept for the duration of the 7 day experiment at sub-confluent density (ideally <90%) would be much more informative in this case, and also help to understand the dynamics within the timecourse. For example, if initially there is cell cycle arrest (at day1, as shown in Fig. 1d), then proliferation rates should reflect that.

See next point.

1. On a more general note: What is the confluence of the 4-7day experiments? Initial density can change the cell's behavior not only for RPE cells (as shown in fig. 7e), but HEK cells are sensitive to that as well. It is critical that experiments for translation, protein content, cell size, etc. be done in sub-confluent conditions, as the over-confluency alone could be a confounder for cell size, translation rates, etc. If this is indeed the way it was done, this should be clarified. Otherwise, this is a critical confounder which should be eliminated.

The risk of the confounding effects of overcrowding is indeed an important point, which we avoided, unfortunately without explicitly mentioning it in the manuscript (assuming that it went without saying). While we had already mentioned the seeding density and type of plate in Materials and Methods, we now address it explicitly both with additional data (new figure supplement 1B) and clarifying additions in the text. In our experience, the most common problem with confluent plates is not that cells grow on top of each other, but that they come off the plate and die. Regarding the cell cycle analysis of Fig. 1D and the proliferation assays of Fig. 1G, note that in the latter, we standardized cell numbers to those of day 1.

1. The speculations about the link to aging and senescence are very interesting, however since these are only hypothesis at this stage, the current phrasing in the abstract is a bit misleading. In fact, I was expecting at least one experiment to deal with aging/senescence, primed by the abstract.

You are perfectly correct. We have now added new experimental evidence that shows cells display activity of the senscence marker SA-βgal after prolonged chronic stress (Figure 10). Please see our response to point #6 of reviewer #1 for further comments.

1. Fig. 2D - nuclei are also getting much larger - what is the contribution of the nuclear increase to the overall cell increase? Does it scale linearly? Or does it contribute more/less compared to the entire cell?

Good point! We now include additional data on nuclear size in Figure 2E and figure supplement 2D, and corresponding additions in Results and Discussion. And as you correctly spotted, nuclei become bigger, too. The data suggest that the ratio of cytoplasm to nuclear size is more or less maintained. One can speculate that nuclei are larger because of partial "unfolding" (opening) of chromatin, which might very well be driven by the activation of Hsf1. But that's for future studies to figure out.

1. Fig 3a-c: in fig. 2a it looks like the knockout of one isoform leads to a basal increase in the expression of the other. However, since different antibodies are used for alpha and beta, the question of whether this increase leads to complete compensation of the total levels of hsp90 cannot be answered. qPCR for common regions could help answer this question, and this could help explain the increased hsf1 activity in the knockouts.

As pointed out in response to reviewer #1, point #4, we had previously determined total Hsp90 levels quantitatively by mass spec (Bhattacharya et al., 2022; see Figure S8 there), and we now mention that explicitly. Moreover, we have now added new data including the quantitation of total Hsp90 levels in Figure 9. RT-PCR might not be of much help considering that we had shown in Bhattacharya et al. 2022 that below-threshold Hsp90 levels (even less than what happens here) trigger translation through an IRES in the Hsp90β mRNA, whose levels don't change.

1. What is the HSE-luc construct used for the hsf1 activity? Is that an artificial HSE? Or the Hspa7 promoter? It would be interesting to check the activity with respect to the hsp90 promoter using a similar assay, to understand whether cells compensation for overall reduction in hsp90 levels is the primary "goal" for hsf1 activation.

The HSE-luc reporter is an artificial construct (we now clarify this in the Materials and Methods). Although Hsp90 is important, Hsf1's goal in life goes well beyond it. It notably also regulates lots of genes in the absence of stress, notably in cancer cells. Fig. 4B is an example of a blot that shows that chronic stress does not dramatically affect the levels of Hsp90α/β.

1. The proteomics data are very interesting, however additional details are missing and it is hard to extract them from source data 1. Specifically - focusing on the 2 hsp90s, what do they look like? The compensation questions above could be answered using the proteomics data as well.

As mentioned above in response to this reviewer's point #5 (and #4 of reviewer #1), we have previously addressed that in a paper that was focused on precisely this issue, and we have adapted the current manuscript accordingly.

1. How many proteins go up/down in the proteomics data? How does this compare between WT and knockout cells? The authors should detail the specific differences, which pathways? Which proteins? otherwise the volcano plots alone, on their own, are really not informative.

We have now added a GO analysis (Figure 5C), and heat maps for chaperones/co-chaperones and Hsp90 interactors (new figure supplements 4 and 5). We have still left some volcano plots because they are a good visualization of the overall changes. The text has been revised accordingly, notably also to clarify what we are trying to show with volcano plots (GO analysis and heat maps).

1. Fig. 3f: cells with hsf1 knockdown even decreased in size after HS. Is this significant? Why could that be?

The be honest, we do not know. A wild speculation would be that Hsf1 is not only required to drive the cell size increase, but that a certain minimal level of Hsf1 is required to maintain normal cell size (specifically in A549 cells?).

1. The siHSF1 cells showing no change in cell size is central to the paper's claims. This should be done in HEK293 cells at least, for which much of the data in the paper is shown, preferably also in RPE1 cells.

We have now added new data with the results obtained with HEK293T cells (Fig. 3F).

1. Technical note: it is very strange that MAFs can be transfected for luciferase assay. Such primary cells, to my knowledge, are largely non-transfectable. How was transfection performed in these cells? The authors should show that these cells can be transfected using imaging, or give a reference.

We did both. We gave references and the experimental details in Materials and Methods, but we now say it even more explicitly in there. Note that the transfection efficiency is not so critical in luciferase assays as one only reads out the activities of the transfected cell population.

1. The claim that proteostasis remains intact and the complexity of the proteome is unchanged should be examined more quantitatively. Specifically, analysis directly comparing between WT and KO cells should be performed: are the induced and repressed proteins the same? Is there a correlation between the levels of significantly changed proteins between WT and KO cells? This analysis should be done for chaperones, hsp90 interactors, as well as for the total proteome. Additionally, proteins whose levels differ could suggest (additional) mechanisms underlying the effects.

This comment also relates back to point #8. We hope that our newly added comment in the Results section associated with the new heat maps makes it clearer what purpose the proteomic data serve and that it is beyond the scope of this paper to quantitate differences further or to home in on this or that protein (with the exception of those proteins we have done immunoblots for). To go deeper into mechanisms is going to be a full project(s) in itself.

1. "Surprisingly, we found that Hsp90α/β KO cells do even better than WT cells under basal conditions (37{degree sign} C) (Figure 4D)." This is not so surprising, in light of the fact that HSF1 activity in these cells is higher, thus their chaperoning capacity should be better (for example, more HSP70 present?), as the authors themselves point out later in the text.

It is surprising considering that there is less of a major molecular chaperone. It's definitely not the first thing you suspect when you knock out Hsp90. But to avoid confusion, we have taken out "surprisingly" and reworded the statement.

1. "Similarly, Hsp90α/β KO cells might do better than WT cells under chronic HSbecause of their ability to further increase the levels of other molecular chaperones, such as Hsp27, Hsp40, and Hsp70, during chronic HS." This relates to the point above - the authors can directly quantify the changes in the levels of all other chaperones, since they have the proteomics data, and substantiate these claims, which are now only suggestions.

The subordinate clause ("... because...") is not a speculation, it is a statement based on the data (Fig. 4B and figure supplement 4A-B, and yes, of course, the proteomic data). However, that KOs indeed do better because of that remains to be proven (hence, the "might do better").

1. In A549 cells, knockout of Hsp90 led to lower basal diffusion coefficient (proxy for cytosolic density) at normal temperatures. Then, at 40 degrees, it seems that the coefficient goes back to being more or less equal to that of WT cells (fig. S5D). How can the authors explain this?

One cannot really compare them one on one. After all, the Hsp90 KOs are different cell lines, their EGFP expression levels may differ, and their heat sensitivity definitely differs. What can be compared is cells of a given cell line (i.e. WT or KO), transfected as a pool and then split to be cultured at different temperatures.

1. P-eIF2alpha and other translation marker western blots should be repeated and quantified and in also performed in A549 KO cells. The latter is very important, as the changed in A549 WT cells during adaptation of all translation regulatory markers: p-eIF2alpha, p-mTOR, and most strikingly total mTOR, are sky-rocketing, while in HEK cells these remain constant. As mTOR is a well-known regulator of cell size, and a target of Hsp90, could it be the major mediator of this effect in A549 cells? And if so, what is the substitute in HEK cells?

We now include bar graphs with quantitation of multiple experiments for both HEK and A549 cells, including for the KOs (Figure 6C-D - figure supplement 8). What they show is that p-mTOR levels increase during chronic stress. But since overall it also increases in Hsp90α/β KO cells, we had to conclude that this cannot explain the differences between cells of different genotypes. We have added a statement to that effect in the corresponding Results section.

1. Figs. 5D (and S5F) are both for HEK cells, while Fig. 5H is for A549. The corresponding plots for both cell lines should be provided for clarity, as the magnitudes in 5D and S5F seem much larger in HEK cells than seen in 5H. If there are differences between the cell lines these should be pointed out, as currently, showing some figures for one and not the other is confusing.

HEK and A549 cells in these experiments, which are different, serve different purposes. We now explicitly mention already in the text of the Results, which cell line is used. Hopefully that makes it less confusing.

1. Fig. 6C lacks a pvalue.

It's missing because it cannot be calculated. The graph shows the average of "only" 2 biologically independent samples (as stated in the legend).

1. Fig. S6C - the legend doesn't match the figure. Additionally, #aggregates should be normalized to the respective #of cells in each micrograph, and p-values should be presented for those normalized values.

For what is now figure supplement 9C, this has been fixed as suggested.

1. Also, under non-HS conditions, Hsp90 knockout cells show less aggregates than the WT. Is this significant (numbers are small, so perhaps it isn't)? What does this mean for the basal proteostasis state of Hsp90 knockout cells? Is it perhaps better than that of the WT?

The suggested way of quantitating the aggregates took care of that. There is no clear difference anymore between WT and KO, but clearly many more aggregates under chronic stress (figure supplement 9C).

1. The data on the connection between size and survival under chronic stress is highly compelling, even though correlative. The authors speculate in the discussion about one possible explanation to the question of how the enlarged size protect from the chronic stress. In fact, their proteomics dataset has the potential to help address, at least in part, their hypothesis about thresholds of certain proteins, by saying which proteins cross the detectability threshold in the data, and which processes these relate to.

What the proteomic data say is that most things don't change (standardized to total protein). While it is possible that a few proteins do change in interesting ways, characterizing those is beyond the scope of this study.

1. Fig. 7G should have a respective quantification with a p-value.

We have added additional data. What is now Fig. 9 shows the quantitation of multiple biological replicates (with p-values).

Minor comments:1. "it is known that acute HS causes ribosomal dissociation from mRNA, which results in atranslational pause (Shalgi et al., 2013)." - This paper showed that acute HS causes ribosomal pausing on mRNAs, not ribosomal dissociation.

We corrected this.

1. Fig. 7E - size bar is missing.

It was actually there, but hard to see. We have improved that in what is now Fig. 8E (and it is now also mentioned in the legend).

**Reviewer #3:**
My main points are outlined in the Public Review. Only a few additional comments are included here:1. The manuscript is quite long and there are places where it could be shortened and tightened for clarity. I'd recommend going through carefully and trying to shorten to improve readability.

We hope that our revisions to address all of the reviewers' comments (and to accommodate more data) make the text more readable. But to make it shorter would have come at the expense of clarity.

1. It wasn't clear to me that the increased luciferase folding in HSP90 KO lines was surprising. It is demonstrated that knockdown of these isoforms can activate HSF1, which increases many chaperones known to promote luciferase refolding.

We address this point in our response to point #13 of reviewer #2 (basically: we took out "surprisingly").

1. Along the same lines. HSP90 knockdown activates HSF1, but doesn't induce basal cell size. However, exogenous overexpression of HSF1 or activation of HSF1 with capsaicin increase cell size. Why are similar things not observed for HSP90 knockdown? Is it the extent of HSF1 activation? This seems a bit unlikely because it looked like activation was similar in KO and capsaicin treated cells.

This must be due to the specifics of these different assays. The levels of Hsf1 protein and activity, and the time course of Hsf1 activity may be different. Moreover, it is likely that the reporter gene readout does not accurately report on all Hsf1 activities at a genome-wide scale.

1. As noted above, does HSP90 depletion impact ISR signaling induced by other types of stress (e.g., ER or mitochondrial stress). Specifically, do you see sustained translational attenuation (and eIF2a phosphorylation) when HSP90 is depleted under these conditions. In other words, does HSP90 have a specific role in globally resolving eIF2a phosphorylation as part of the ISR or is that specific to certain types of stress.

Although we now include data to show that tunicamycin (and therefore presumably the UPR/ISR) also induces a cell size increase, comprehensively analyzing what we refer to as RSR across different types of stresses (including mitochondrial and ER stresses) in the background of different Hsp90 genotypes and cell lines goes well beyond the scope of the current study.